Corrected: Author correction

# Recurrent acquisition of cytosine methyltransferases into eukaryotic retrotransposons

Alex de Mendoza[1,2], Amandine Bonnet[3], Dulce B. Vargas-Landin[1,2], Nanjing Ji[4], Hongfei Li[4], Feng Yang[4], Ling Li[4], Koichi Hori[5], Jahnvi Pflueger[1,2], Sam Buckberry [1,2], Hiroyuki Ohta[5], Nedeljka Rosic[6,7], Pascale Lesage [3], Senjie Lin [4,8,9] & Ryan Lister [1,2]

Transposable elements are in a constant arms race with the silencing mechanisms of their host genomes. One silencing mechanism commonly used by many eukaryotes is dependent on cytosine methylation, a covalent modification of DNA deposited by C5 cytosine methyltransferases (DNMTs). Here, we report how two distantly related eukaryotic lineages, dinoflagellates and charophytes, have independently incorporated DNMTs into the coding regions of distinct retrotransposon classes. Concomitantly, we show that dinoflagellates of the genus *Symbiodinium* have evolved cytosine methylation patterns unlike any other eukaryote, with most of the genome methylated at CG dinucleotides. Finally, we demonstrate the ability of retrotransposon DNMTs to methylate CGs de novo, suggesting that retrotransposons could self-methylate retrotranscribed DNA. Together, this is an example of how retrotransposons incorporate host-derived genes involved in DNA methylation. In some cases, this event could have implications for the composition and regulation of the host epigenomic environment.

[1] Australian Research Council Centre of Excellence in Plant Energy Biology, School of Molecular Sciences, The University of Western Australia, Perth, WA 6009, Australia. [2] Harry Perkins Institute of Medical Research, Perth, WA 6009, Australia. [3] Université Paris Diderot, Sorbonne Paris Cité, INSERM U944, CNRS UMR 7212, Institut Universitaire d'Hématologie, Hôpital St. Louis, 75010 Paris, France. [4] State Key Laboratory of Marine Environmental Science and College of Ocean and Earth Sciences, Xiamen University, 361102 Xiamen China. [5] School of Life Science and Technology, Tokyo Institute of Technology, Yokohama City, Kanagawa 226-8501, Japan. [6] School of Health and Human Sciences, Southern Cross University, Gold Coast, QLD 4072, Australia. [7] School of Biological Sciences, The University of Queensland, St. Lucia, QLD 4225, Australia. [8] Institute of Systems Genomics, University of Connecticut, Storrs, CT 06269, USA. [9] Department of Marine Sciences, University of Connecticut, Groton, CT 06340, USA. Correspondence and requests for materials should be addressed to A.dM.(email: alex.demendoza@uwa.edu.au) or to R.L. (email: ryan.lister@uwa.edu.au)

Genomic cytosine methylation (mC) has been shown to transcriptionally silence repetitive elements in many eukaryotes, including plants, algae, fungi and vertebrates. Given the disparity of lineages that share this trait, mC-dependent transposable element silencing has been suggested as one of the original roles of mC in eukaryotes, together with active gene body methylation[1, 2]. The enzymes responsible for mC deposition, the cytosine methyltransferases (from here on referred to as DNMTs), are diverse and ancient with many families tracing their origin to the last common eukaryotic ancestor[2, 3]. Distinct DNMT families can methylate different substrates (DNA or RNA) and in different local sequence contexts, predominantly at CG dinucleotides but also at CH (H=A, T or C) sites[4]. The sequence context of cytosine methylation has different regulatory outcomes in a lineage-specific manner. For instance, CHH and CHG trinucleotides mark plant transposable elements, while CG methylation is found on active gene bodies and on transposable elements[1]. In mammals, CG methylation is widespread but absent in most promoters and active regulatory elements, while CH methylation is anticorrelated with gene transcription and found in a restricted range of cell types[5].

Ancient coexistence with DNMTs prompted transposable elements to evolve ways to escape from being targeted and subsequently methylated in order to remain active in host genomes[6]. Counterintuitively, two studies reported rare cases of cytosine DNMTs linked to reverse transcriptase domains[3, 7], but did not determine whether these DNMTs belong to transposable elements. Here, we characterise how cytosine-specific DNMTs have been incorporated into distinct classes of retrotransposons independently. Moreover, we assess their functional activity and profile the epigenomic characteristics of their algal hosts that belong to the charophyte and dinoflagellate lineages.

## Results

**Symbiodinium has hundreds of DNMTs within retrotransposons.** One of the most recent reports from this alleged DNMT-retrotransposon association comes from the genome of the dinoflagellate Symbiodinium kawagutii[7], a member of a genus that plays a key role as coral symbionts[8, 9]. To confirm and characterise the structure of the DNMT-encoding genes, we used a combined genome annotation approach based on RNA-seq and ab initio intronless gene models, revealing that DNMT's in Symbiodinium species are two orders of magnitude more abundant in copy number than observed in any other eukaryote (Fig. 1a). Some DNMT-encoding genes are archetypal Symbiodinium genes[10]: very long (S. kawagutii median 11.2 kb, S. minutum 8.2 kb), with short exons (median S. kawagutii 65 bp, S. minutum 68 bp), and in a head-to-tail orientation with surrounding genes (Fig. 1b). However, the vast majority of DNMTs are found in intronless genes (>90%). Exon-rich DNMT-encoding genes also show higher transcript levels compared to the intronless DNMTs, the majority of which are silent (Fig. 1c).

To assess the diversity of Symbiodinium DNMTs, we classified them into the established eukaryotic DNMT classes[11]. Using phylogenetic reconstruction of all Symbiodinium DNMT exon-rich genes, we could not find members of the well-characterised Dnmt1 and Dnmt3 families, responsible for maintenance and de novo methyltransferase activity in other eukaryotes[4]. However, we identified several paralogues of Dnmt5, Dnmt6 and the tRNA methyltransferase Dnmt2 families (Fig. 1d, f). Unlike in other eukaryotes, most of these Symbiodinium DNMTs do not encode for any additional known protein domains (Supplementary Fig. 1). By screening the available transcriptomes of other dinoflagellates, we found that this repertoire of expressed DNMTs is not unique to Symbiodinium, but rather is relatively well conserved across the lineage (Supplementary Fig. 2). Interestingly, two dinoflagellate species express Dnmt3 orthologs (Fig. 1f and Supplementary Fig. 2), which indicates that Dnmt3 was ancestrally present in dinoflagellates. These two Dnmt3 transcripts lack additional domains beside the DNMT. Together, this reveals that dinoflagellates express a set of DNMTs unlike that of any other eukaryote, simplified in protein domain architecture but diversified in copy number.

In contrast to exon-rich DNMTs, Symbiodinium intronless DNMTs belong either to a Dnmt3 family or to an orphan lineage of Symbiodinium-specific DNMTs. The expanded intronless DNMT families are not retrocopies of transcribed genes, but are embedded in retrotransposon coding regions. The Symbiodinium-specific DNMTs are mostly found in an open reading frame (ORF) upstream of another ORF encoding a reverse transcriptase (Fig. 1e, SymbioLINE-Dnmts). Their architecture is typical of long interspersed nuclear element (LINE) retrotransposons, but no LINEs described to date encode a DNMT in its ORF1[12]. In contrast, the Dnmt3 DNMTs are found in retrotransposons structurally similar to DIRS (Dictyostelium intermediate repeat-sequence) retrotransposons (Fig. 1e, SymbioDIRS-Dnmt3). Previously, adenine methyltransferases similar to bacterial Dam N6A methylases had been identified in retrotransposons of the DIRS class[13–15]. However, Dam-like adenine methyltransferases are structurally unrelated to cytosine DNMTs and are rarely encoded by native eukaryotic genes[3]. Unlike LINEs, DIRS retrotransposons have two long terminal repeats (LTRs) flanking the coding sequences[13]. In SymbioDIRS-Dnmt3, the DNMT domains are found in the Pol polyprotein, sharing an ORF with the reverse transcriptase. A separate ORF encodes a tyrosine recombinase characteristic of DIRS retrotransposons, responsible for the integration of the retrotranscribed cDNA into the genome. Surprisingly, two distinct SymbioDIRS-Dnmt3 structures coexisting in Symbiodinium genomes, which differ by having distinct orders of domains within the Pol polyprotein and shuffled ordering of the integrase and the Pol ORFs. Notably, all three structurally distinct families of Symbiodinium retrotransposons encode functionally conserved key amino acid positions in the DNMT (Supplementary Fig. 3). SymbioDIRS-Dnmt3 sequences cluster within the clade of eukaryotic Dnmt3 (Fig. 1f) together with two transcripts from other dinoflagellate species, supporting their eukaryotic origins as they do not cluster with viral or bacterial sequences. However, a yet to be sequenced virus might have been the origin of the DNMTs from both SymbioLINE-Dnmt and SymbioDIRS-Dnmt3, as DNA transposons encoding a DNMT have been identified in marine viruses[16]. Nevertheless, given the current evidence, our data suggest that native Dnmt3 genes were lost after being acquired by retrotransposons in the lineage that gave rise to Symbiodinium.

**Symbiodinium DNMT-retrotransposons are old and active.** To understand the evolutionary dynamics of these unique retrotransposons within Symbiodinium genomes, we constructed phylogenetic trees using the reverse transcriptase domain sequences. This revealed that many SymbioLINE-Dnmt clades feature sequences from all four Symbiodinium species, while others are species-specific, indicating that the common ancestor of the Symbiodinium genus already possessed various members of this retrotransposon family (Fig. 2a). Additionally, we observe complete copies in all major clades, indicating this is not the result of a recent massive burst of replication of a single copy, but rather many distantly related subfamilies being retro-transcriptionally active. In the case of SymbioDIRS-Dnmt3, we observe a similar pattern, with sequences from all species being intermingled within various clades (Fig. 2b), suggesting ancestral

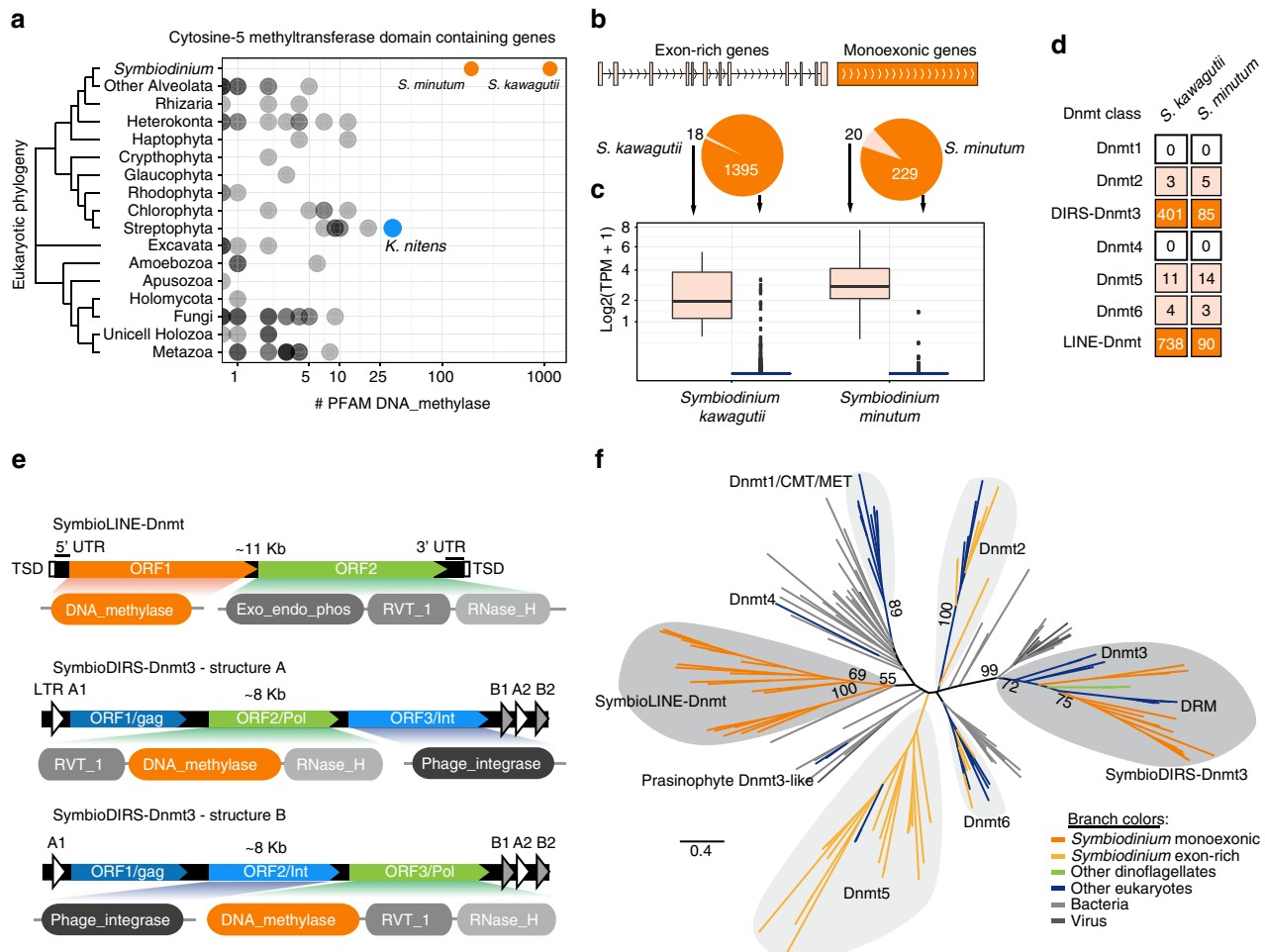

**Fig. 1** *Symbiodinium* genomes encode hundreds of DNMTs. **a** Number of DNMT genes encoded in eukaryotic lineages based on a data set of 96 genomes (Supplementary Data 1). Each dot corresponds to the number of genes containing a DNMT domain (PF00145, DNA_methylase) in a given genome. **b** Proportion of distinct DNMT gene architectures in *Symbiodinium*, pale shade for exon-rich, darker shade for monoexonic. **c** Transcriptional level of DNMT-encoding genes according to gene architecture in *Symbiodinium*. Boxplot centre lines are medians, box limits are quartiles 1 (Q1) and 3 (Q3), whiskers are 1.5 × interquartile range (IQR) and points are outliers. **d** Classification of DNMTs in *Symbiodinium* genomes, showing number of copies per family. Colour code reflects the type of gene architecture for each family: pale shade = exon-rich, darker shade = monoexonic. **e** Structure of *Symbiodinium* retrotransposons that encode for DNMTs and domain architectures of the open reading frames (ORFs). Domains named as in Pfam searches: Exo_endo_phos (PF03372) is an endonuclease domain, RVT_1(PF00078) is a reverse transcriptase, RNase_H (PF00075) is a RNase type H and Phage Integrase (PF00589) is a tyrosine recombinase. SymbioLINE-Dnmt show flanking tandem site duplications (TSD), conserved 5′ and 3′ UTR regions and lack Poly-A tails. SymbioDIRS-Dnmt3 show two sets of LTRs (represented by triangles A and B) in an arrangement typical of DIRS retrotransposons: A-coding region-B-A-B. **f** Maximum likelihood phylogenetic tree of a representative group of eukaryotic, bacterial and *Symbiodinium* DNMTs. Hundred parametric bootstrap replicates are shown as nodal supports for selected groups. Colour code indicates sequence taxonomic affiliation of each branch as shown in legend

diversity. Moreover, we find that the two different architectures of SymbioDIRS-Dnmt3 are distantly related, indicating that both retrotransposons have independently acquired the Dnmt3 domain into their *Pol* ORF. Comparing the reverse transcriptase phylogenies to the phylogenetic tree of the DNMT domain, we observe congruent topologies (Supplementary Fig. 4), indicating that the acquisition of DNMT domains occurred early on in the evolution of both SymbioDIRS-Dnmt3 retrotransposon structures. Thus, the three *Symbiodinium* DNMT-bearing retrotransposon families are ancestral and show multiple active subfamilies.

*Symbiodinium* genomes encode other transposable elements besides SymbioLINE-Dnmt and SymbioDIRS-Dnmt3 (Fig. 2c and Supplementary Fig. 5). For example, there are DIRS and LINE elements lacking DNMTs (Fig. 2c). Therefore, ancestral forms of both families co-exist with their DNMT version. Notably, the frequency of LINE retrotransposons in *Symbiodinium* genomes is

similar to that observed in some vertebrates[17], which likely facilitated the ancestral co-option of the DNMT domain. However, the most abundant retrotransposons in *Symbiodinium* genomes belong to the LTR Copia superfamily[18], which lacks any DNMT domain. When restricting the search for recent copies, all types of retrotransposons show similar expansion dynamics, some new and actively expanding lineages but a majority of divergent lineages (Supplementary Fig. 6a, b). Furthermore, analysing the expansion of specific subfamilies of DNMT and non-DNMT encoding retrotransposons, we found that both showed similar patterns of expansion (Supplementary Fig. 6c). This indicates that encoding a DNMT is not required in order to be retro-transcriptionally active in *Symbiodinium* genomes. All transposable elements show similar insertion distribution in *S. kawagutii*, mostly found in introns and intergenic regions and distributed across many scaffolds (Supplementary Fig. 7a, b).

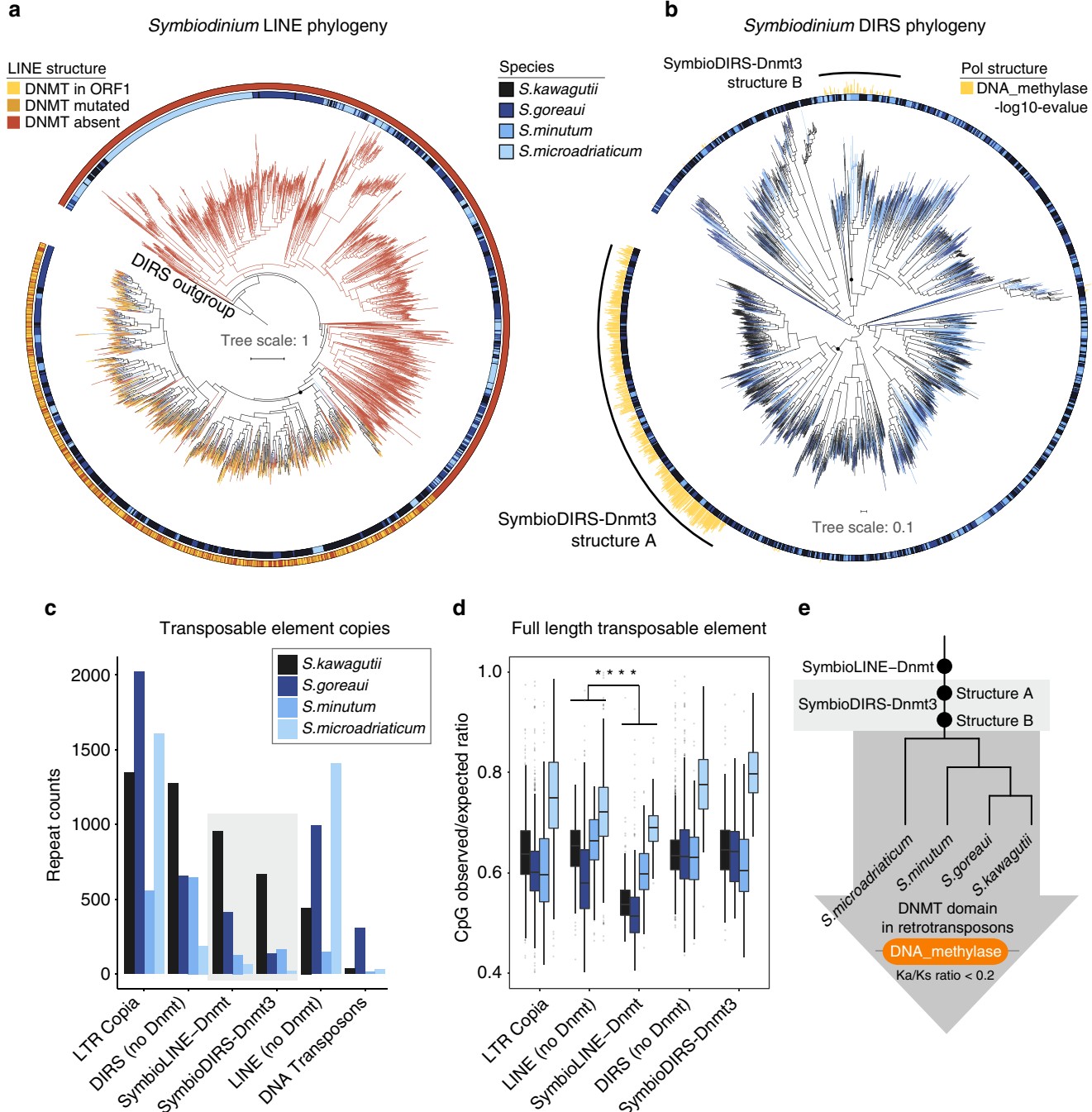

**Fig. 2** DNMT-retrotransposon evolutionary dynamics in *Symbiodinium*. **a** Maximum likelihood phylogenetic tree based on the ORF2 of *Symbiodinium* LINE retrotransposons, rooted using DIRS reverse transcriptases as an outgroup. The internal circle indicates species affiliation of each sequence and the outer circle indicates the architecture of the LINE element. DNMT mutated indicates that there are premature stop codons in ORF1 or ORF2. DNMT absent implies that there is no evidence of upstream ORF1 encoding a DNMT. **b** Maximum likelihood phylogenetic tree based on *Pol* ORF (retrotranscriptase and RNase_H domains) of DIRS-like retrotransposons. The inner circle indicates species affiliation and the barplot indicates the degree of DNMT conservation based on a hmmsearch alignment *e*-value. **c** Number of transposable element copies in each *Symbiodinium* species. **d** Comparison of the CG observed/expected ratio across all *Symbiodinium* retrotransposon types. SymbioLINE-DNMTs have fewer CGs than the rest (asterisks represent Wilcoxon one-sided rank-sum test $p < 0.01$) of retrotransposons in all four species, while SymbioDIRS-Dnmt do not show a significant difference when compared to DIRS elements without DNMT ($p > 0.01$). Boxplot centre lines are medians, box limits are quartiles 1 (Q1) and 3 (Q3), whiskers are 1.5 × interquartile range (IQR) and points are outliers. **e** Schematic species tree of *Symbiodinium* species showing the evolutionary origins of DNMT encoding retrotransposons in this lineage and the subsequent conservation of DNMT domain as shown in Supplementary Figs. 8 and 9. Phylogenetic relationships between *Symbiodinium* species is based on previous publications[24, 38]

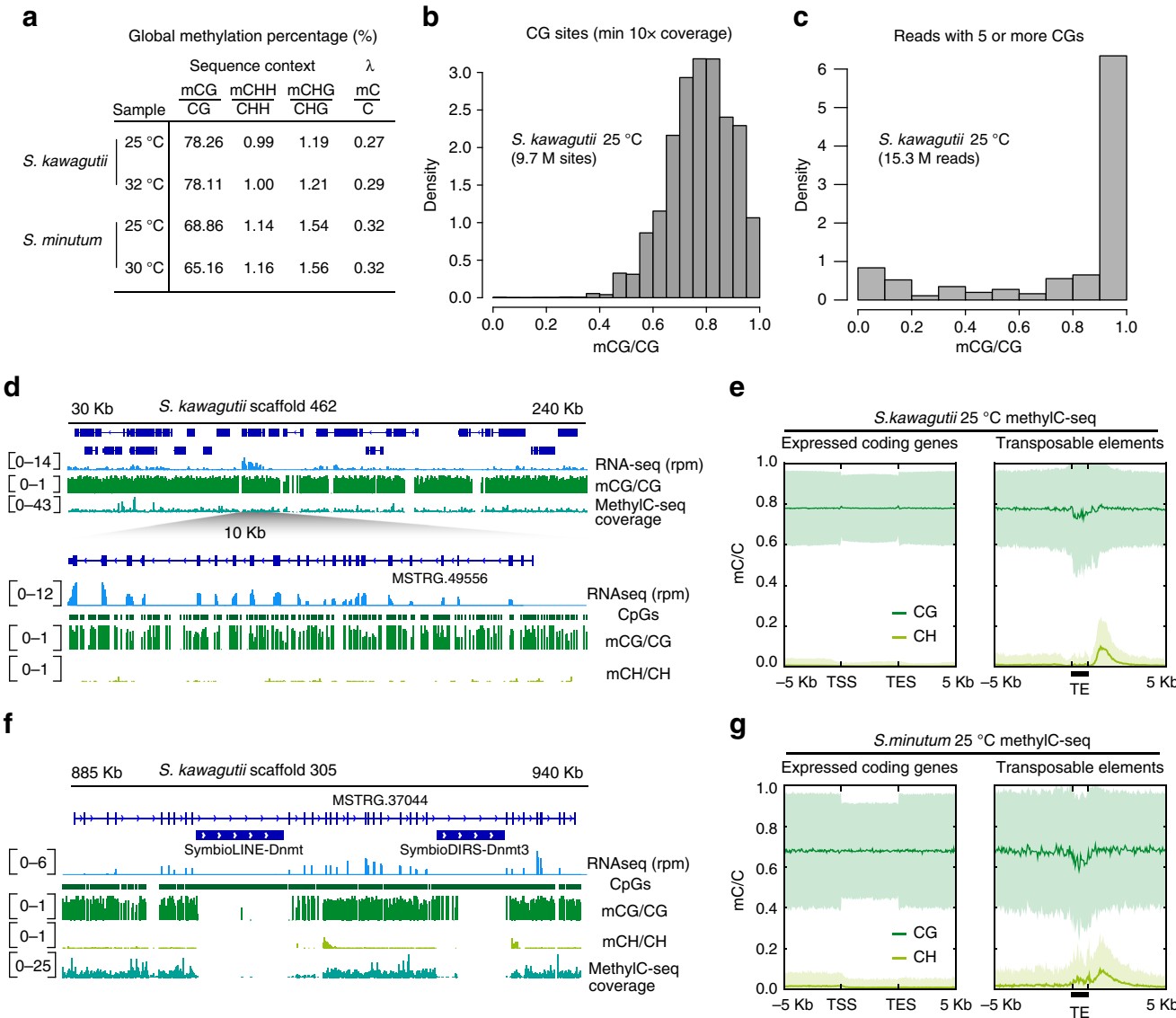

**Fig. 3** Widespread CG and localised CH methylation in *Symbiodinium*. **a** Percentage of methylated cytosines in distinct sequence contexts in four MethylC-seq libraries. H represents any base but a G. The lambda genome methylation indicates the bisulfite non-conversion rate for each library. **b** Unimodal distribution of methylation levels (the ratio of C base calls to total base calls) on individual CG sites with high coverage (≥10 reads) in *S. kawagutii* 25 °C culture sample. **c** Methylation level of mapped reads with ≥5 CGs in *S. kawagutii* 25 °C culture sample. **d** Genome browser display showing the profile of methylation on genes expressed at different levels in a single scaffold. rpm reads per million. **e** Methylation profiles in CG and CH contexts on *S. kawagutii* genes and transposable elements. Mean (thick line) and standard deviation (shade) methylation levels for each relative position around genomic features. **f** Genome browser display demonstrating CH methylation enrichments on 3′ end of silent DNMT encoding retrotransposons found in introns of an expressed gene. **g** Methylation profiles in CG and CH contexts on *S. minutum* genes and transposable elements. Line represents mean and pale shade represents standard deviation. TSS transcriptional start site, TES transcriptional end site, TE transposable element

The recurrent acquisition of DNMTs into retrotransposons suggests a role of this domain during the retrotransposition process, likely involved in methylation of cytosines of the retrotranscribed copy. However, given that methylated cytosines in CG dinucleotides are prone to deamination into thymine, there would be an expected compositional bias against CG dinucleotides in some of these retrotransposons if self-methylation during retrotransposition was occurring. To test this, we calculated the observed versus expected ratio of CG dinucleotides in retrotransposon sequences, revealing that only SymbioLINE-Dnmts show a significantly reduced frequency of CG dinucleotides compared to whole genome or to *Symbiodinium* LINEs lacking DNMT domains (Fig. 2d and Supplementary Fig. 7c, d, e). This observation holds for recently duplicated retrotransposons

(Supplementary Fig. 7c), This suggests that C-to-T increased mutability could be affecting CG dinucleotides in SymbioLINE-Dnmts, indicating that active RNA or cDNA methylation is occurring in SymbioLINE-Dnmt. However, other causes such as biased mismatch repair and specific targeting by host DNMTs could also be responsible for the CG dinucleotide depletion of SymbioLINE-Dnmts.

To test whether the DNMT domain has potentially advantageous roles for the retrotransposons, we calculated the ratio between non-synonymous substitutions (Ka) and synonymous substitutions (Ks) of the DNMT domain for copies of SymbioLINE-Dnmt and SymbioDIRS-Dnmt3 belonging to the same species. In all versus all comparison, a majority of DNMT sequences show significant levels of purifying selection (Ka/Ks <

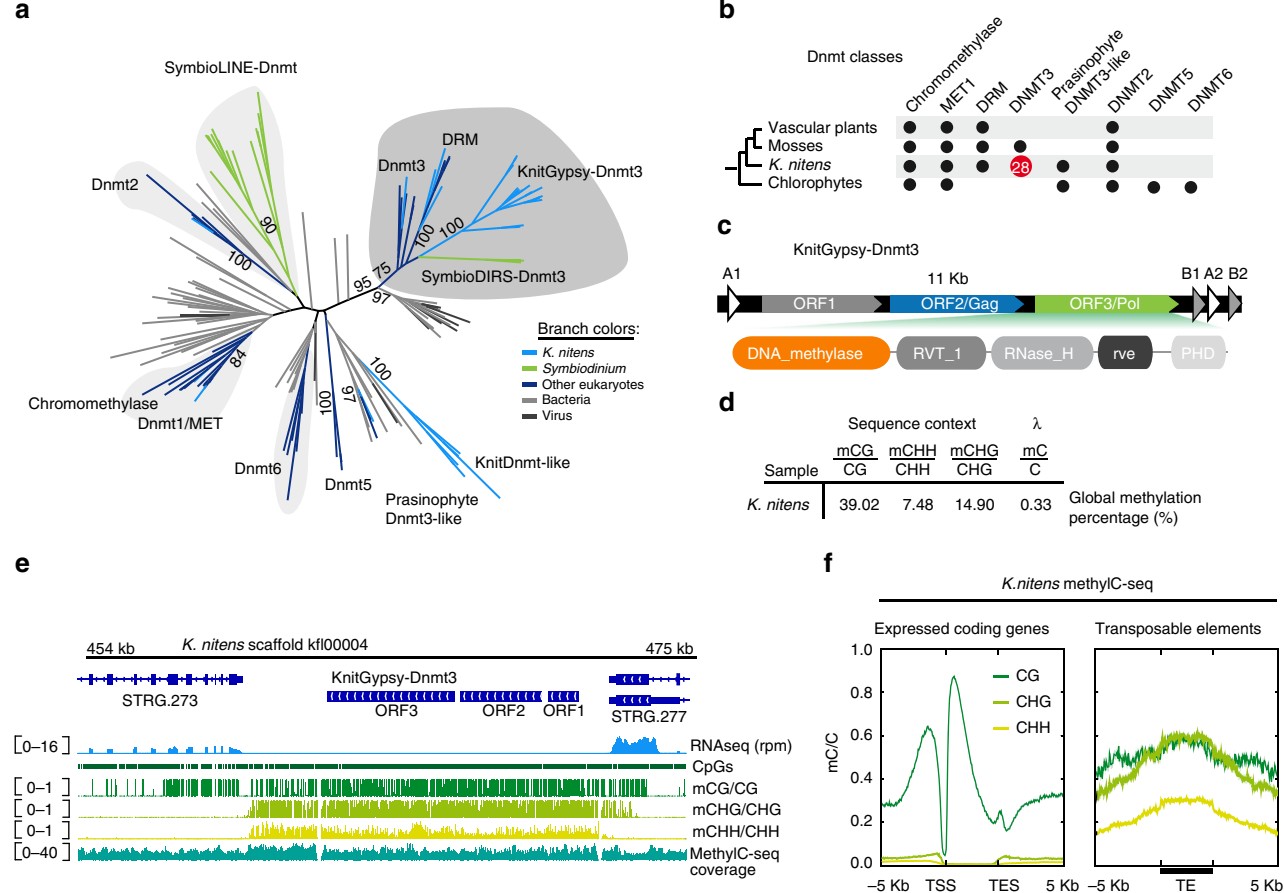

**Fig. 4** Independent Dnmt3 acquisition into *K. nitens* retrotransposons. **a** Maximum likelihood phylogenetic tree of eukaryotic DNMTs including *K. nitens* orthologs. **b** Distribution of DNMT classes in the viridiplantae, in red copy number of Dnmt3 found in *K. nitens*. **c** Structure of *K. nitens* retrotransposons that encode for DNMTs and domain architectures of the ORFs. The structure of repeats is the same as that of a DIRS retrotransposon, but the domain composition is typical of Gypsy retrotransposons. **d** Percentage of methylated cytosines in distinct sequence contexts in *K. nitens* MethylC-seq library. **e** Genome browser display showing the profile of cytosine methylation in CG, CHH and CHG contexts on KnitGypsy-Dnmt3 retrotransposons. rpm reads per million. **f** Methylation profiles in CG, CHH and CHG contexts on *K. nitens* genes and transposable elements. Line represents mean and pale shade represents standard error of the mean. TSS transcriptional start site, TES transcriptional end site, TE transposable element

0.2), usually showing saturated synonymous substitution rates (Supplementary Fig. 8). The Ka/Ks ratios for the reverse transcriptase domain of the same retrotransposons showed similar levels of purifying selection, indicating that both domains are conserved at similar ratios (Supplementary Figs. 8 and 9a). In addition, when comparing retrotransposon DNMT sequences belonging to different lineages to the closest ortholog of another species, the DNMT domain also shows significant purifying selection (Ka/Ks < 0.2) (Supplementary Fig. 9b). This reveals that many lineages of retrotransposons have actively conserved the DNMT domains after diverging from the original copies in the last common ancestor of the *Symbiodinium* genus and within species-specific diversifications (Fig. 2e).

Some of the transposable element families bearing a DNMT are actively transcribed (Supplementary Fig. 10a). Interestingly, the transcription of SymbioDIRS-Dnmt3 is biased towards the 5′ region of the element, not covering the *Pol* or the recombinase ORFs (Supplementary Fig. 10b, d). In contrast, SymbioLINE-Dnmt shows transcription biased towards the 3′ end, overlapping ORF2 and the 3′ UTR (Supplementary Fig. 10c, d). Thus, despite most DNMT retrotransposons being silent, some are still partially transcriptionally active. As non-specific reverse transcriptase activity can give rise to host gene retrocopies[19], similarly, accidental uncontrolled transcription and non-specific activity of some of the hundreds of

retrotransposon DNMTs encoded in the genome could affect the composition of the host methylome.

### *Symbiodinium* features unique patterns of cytosine methylation.
To characterise the composition of the *Symbiodinium* DNA methylomes, we performed whole-genome bisulfite sequencing by MethylC-seq on *S. kawagutii* and *S. minutum*. We sequenced two samples for each species at different temperature conditions, as thermal-related stress is key in the symbiotic relationship to the coral host[8]. Global cytosine methylation levels were similar across samples of the same species, indicating minimal effect of thermal stress on global methylation levels (Fig. 3a). Despite *S. kawagutii* and *S. minutum* exhibiting a 10% difference of global methylation levels, in both species most methylation occurs at CG dinucleotides, at global levels as high as that observed in the mouse or human genome (~70%)[8]. However, in contrast to mammals, the distribution of relative methylation levels at single CG sites is unimodal rather than bimodal, with most CGs showing medium to high methylation (mCG/CG > 0.2), and less than 0.1% of CGs being completely unmethylated compared to ~3.7% in humans (Fig. 3b). At the single read level, most reads are completely methylated, but some have intermediate levels or stretches of unmethylated CGs, indicating a high degree of cell-to-cell heterogeneity (Fig. 3c). Another characteristic of CG methylation in

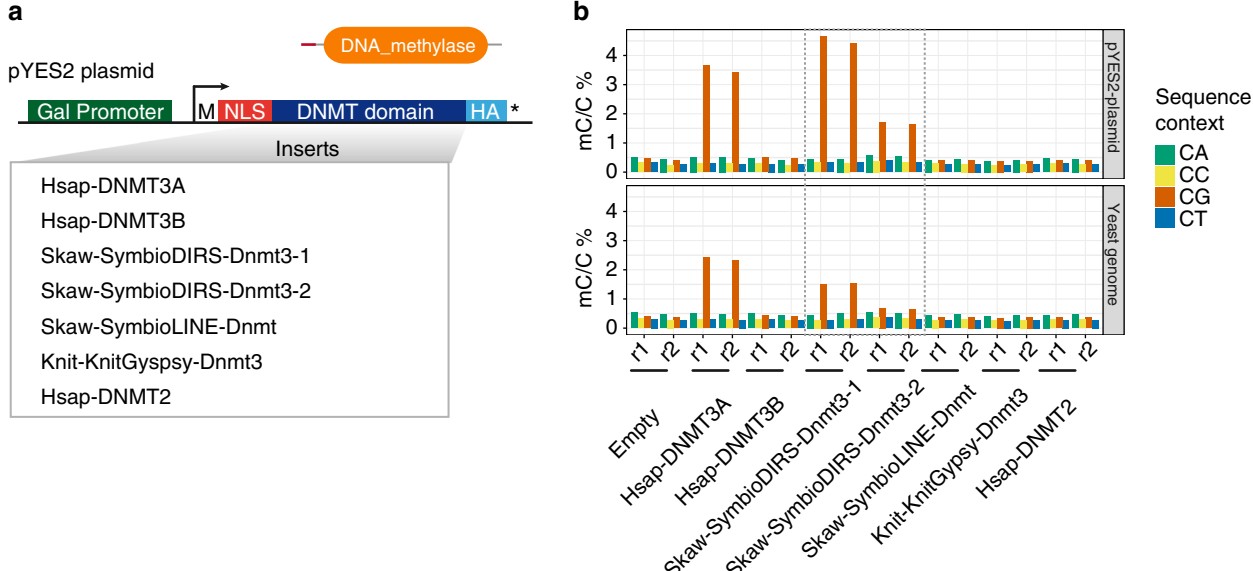

**Fig. 5** De novo methylation of CGs by retrotransposon-encoded DNMTs. **a** Schematic representation of the pYES2 construct and the list of DNMT domain inserts tested. M starting methionine, NLS nuclear localisation signal, HA human influenza hemagglutinin tag, Hsap *Homo sapiens*, Skaw *S. kawagutii*, Knit *K. nitens*. **b** Methylation induction in pYES2-DNMT-expressing yeast lines measured in duplicates (r1 and r2) as a percentage of mC calls against all Cs in each of the four dinucleotide sequence contexts. Independent measures of methylation levels reported for the episomal pYES2 construct and the native yeast genome. Data for the non-conversion for each experiment (spiked-in lambda genome) is in Supplementary Fig. 15b

most eukaryotes is its symmetry[11], where the methylation levels of the cytosine on both strands of the CG are highly correlated. In *Symbiodinium*, we found low correlation of the methylation level between symmetric Cs in CG sites (Supplementary Fig. 11a), strengthening the case for cell heterogeneity. Cell heterogeneity could reflect different replication time points or hidden life cycle phases among cells in culture. Overall, these methylation patterns suggest that hypermethylation is the default state for CG sites, with inherent heterogeneity in single cells at most CG sites. Lack of strict methylation maintenance due to absence of Dnmt1 suggests a non-specific but rapid accumulation of methylation at CGs after genome replication.

In terms of impact on gene regulation, CG methylation is homogeneous across the *Symbiodinium* genome and it does not correlate with transcription (spearman $r = -0.03$). Similar CG methylation levels are observed on genes with distinct expression levels or on transposable elements (Fig. 3d, e, g). Furthermore, there is no depletion of methylation in promoters, which is typical of most eukaryotes with highly methylated genomes (e.g., *Chlorella* or vertebrates)[20, 21]. Testing for localised differentially methylated regions (DMRs) between samples did not identify any significant DMRs, which further strengthens the hypothesis of CG hypermethylation as a default state not being specifically maintained nor locally regulated.

In contrast to CG methylation, methylation in the CH sequence context was low (mCH/CH ~1%) (Fig. 3a). Unlike in plants and other algae[11], there is no difference between CHH and CHG methylation levels, which suggests a lack of specificity towards CHG trinucleotides (Fig. 3a). CH methylation is clearly enriched on transposable elements, biased towards the 3′ ends (Fig. 3e, f, g). When filtering for transcribed transposable elements (>0.5 trancripts per million) with sufficient coverage (mean >2 × per C), we observe that most lack 3′ CH methylation enrichment (10/15), indicating that CH methylation is usually lacking in the few active copies in the *S. kawagutii* genome. Variability of CH methylation levels is also dependant on transposable element class, but independent of the presence of a DNMT in its coding region (Supplementary Fig. 12a, b).

Despite CH methylation levels being typically lower than at CG sites, they are positionally conserved across samples (Supplementary Fig. 11b, c). Thus, CH methylation, despite occurring much less frequently than CG methylation, appears to be primarily targeted to transposable elements.

Together, these results demonstrate that *Symbiodinium* has highly divergent methylation patterns compared to other eukaryotes, which might reflect a convoluted evolutionary history of co-adaptation between the host epigenome and the retrotransposons bearing DNMTs.

**Independent Dnmt3 acquisition in charophyte retrotransposons**. To test if our observations in *Symbiodinium* are unique in eukaryotes, we focused on the third eukaryotic genome encoding higher number of DNMTs in our survey, the charophyte *Klebsormidium nitens* (Fig. 1a). *K. nitens* is a filamentous alga closely related to land plants, which shows many molecular similarities to its land multicellular relatives[22]. To assess the level of conservation of the plant methylation enzymes, we classified the repertoire of DNMTs in *K. nitens* (Fig. 4a, b). As expected, its repertoire is similar to that of land plants, but unexpectedly, *K. nitens* shows an expansion of Dnmt3 DNMTs (Fig. 4a, b), which unlike plant Dnmt3-related domains rearranged methyltransferases (DRM), do not show any rearrangement in the DNMT domain. Although one of the *K. nitens* Dnmt3 enzymes is a multi-exonic gene and clusters with metazoan and bryophyte sequences, the rest of the sequences cluster in a more divergent clade and lack introns. These Dnmt3-related DNMTs are incorporated into the *Pol* polyprotein ORF of a LTR Gypsy retrotransposon (Fig. 4c, KnitGypsy-Dnmt3) and show conserved key functional amino acids (Supplementary Fig. 13a). Moreover, the DNMT domains across distinct copies of KnitGypsy-Dnmt3 show evidence for purifying selection (Ka/Ks < 0.2, Supplementary Fig. 13b), suggesting that they are actively conserved. LTR Gypsy retrotransposons are distantly related to DIRS or LINE retrotransposons (Supplementary Fig. 14a), thus there is no relationship with the *Symbiodinium* retrotransposons. These data

reveal an independent acquisition of Dnmt3-DNMTs by Gypsy retrotransposons in the *K. nitens* lineage.

To test if independent evolution of Dnmt3-encoding retrotransposons could be linked to similarly derived cytosine methylation patterns, we profiled the *K. nitens* methylome. Unlike *Symbiodinium*, the global methylation patterns were similar to that of land plants, with a preference for CG methylation and relatively high levels of CHH and CHG methylation (Fig. 4d). CG methylation was particularly enriched in active gene bodies and excluded from the promoter regions (Fig. 4e, f). As in plants, CG gene body methylation is positively correlated with gene expression level (spearman $r = 0.427$, Supplementary Fig. 14b, c, d). Both CHH and CHG methylation are found on silent transposable elements, for instance on copies of KnitGypsy-Dnmt3 (Fig. 4e), but not in the multi-exonic version of *K. nitens* Dnmt3 (Supplementary Fig. 14e). Although *Symbiodinium* and *K. nitens* have evolved DNMT-encoding retrotransposons, their methylomes have not evolved similar characteristics. However, in both cases, CH methylation appears to be involved in transposable element targeting, while CG methylation is targeting active gene bodies and silent transposons alike. Therefore, the ability of retrotransposons to methylate DNA in a specific sequence context indicates how they may interact with the host epigenome.

**Retrotransposon DNMTs can catalyse de novo CG methylation.** To examine the functional conservation of the retrotransposon DNMTs, we expressed a DNMT from each retrotransposon class in *Saccharomyces cerevisiae*. Budding yeast lacks native cytosine DNA methylation, yet their DNA can be methylated experimentally, and is therefore an ideal model for in vivo methylation[23]. Given that all DNMTs are present in large and diverse multi-domain proteins that could affect their affinity for DNA, we cloned each isolated DNMT catalytic domain fused with a nuclear localisation signal (NLS) and an HA epitope (Fig. 5a). We conditionally expressed the transgenes and isolated the DNA of the cells in the stationary phase (Supplementary Fig. 15a). Profiling the levels of mC in each yeast strain, we confirmed that the negative controls (empty vector and tRNA methyltransferase DNMT2) did not show any methylation (Fig. 5b and Supplementary Fig. 15b). Conversely, the human DNMT3A DNMT domain was able to induce methylation in the CG context, both in the yeast genome and in the episomal plasmid (Fig. 5b). The difference between methylation levels in both DNA compartments might be due to the high copy number and likely distinct chromatin landscape of the episomal plasmids (Supplementary Fig. 15c). Among the tested DNMTs, we could detect induced de novo methylation by DNMTs of the SymbioDIRS-Dnmt3 retrotransposons (Fig. 5b). Interestingly, both distantly related copies have the ability to methylate de novo CGs. Both SymbioDIRS-Dnmt3 copies and DNMT3A methylate CG asymmetrically (Supplementary Fig. 15d), and methylated sites do not show any sequence preference besides the CG dinucleotide (Supplementary Fig. 15e). The lack of de novo methylation observed for the SymbioLINE-Dnmt and KnitGyspy-Dnmt3 DNMTs is inconclusive, given that, despite being translated (Supplementary Fig. 15f), the human DNMT3B was also unable to induce de novo methylation. Lack of required host-specific co-factors, improper folding or incorrect subcellular localisation could explain the lack of activity of those DNMTs. Together, this demonstrates that some retrotransposon-encoded DNMTs are functionally active on DNA and can methylate CG dinucleotides.

**Discussion**
Here, we report an unprecedented example of divergent retrotransposons that have recruited host genes involved in cytosine methylation in two distantly related lineages of eukaryotes. The

origin of these retrotransposons could be linked to the host-specific patterns of DNA methylation. In the striking case of *Symbiodinium*, given the demonstrated ability of SymbioDIRS-Dnmt3 DNMTs to methylate CGs and the pervasive transcription of these retrotransposons, it is reasonable to speculate that ubiquitous CG methylation in the *Symbiodinium* genome could be a response of the host to uncontrolled retrotransposon DNMT methylation. If locally restricted regions of the genome would require lack of methylation for proper regulation, such as most eukaryotic promoters, undesired retrotransposon-derived methylation, and its maintenance through Dnmt1, could have detrimental effects on the host fitness. This could explain why some basic functions of cytosine methylation have been lost in this lineage through methylation of all CGs as the default state, which would represent an inverse case to the loss of DNA methylation observed in some other species such as *Drosophila* and *C. elegans*. However, the methylation patterns observed in *Symbiodinium* genomes are likely deposited by the several copies of highly expressed exon-rich DNMTs belonging to Dnmt5 and Dnmt6 families, as in other algal lineages were Dnmt1 and Dnmt3 are absent[11]. On the other hand, dinoflagellates are known for their derivate genomic characteristics such as pervasive trans-splicing, extreme chromatin condensation and rarity of canonical nucleosomes, which could reinforce the evolution of such a non-canonical methylome[8, 9]. New genomic and epigenomic data from other dinoflagellate species should enable establishment of whether the transposons evolved before or after the particular methylome characteristics originated. Finally, the evolution of DNMT-encoding retrotransposons might be unrelated to the host methylome composition, for example, *K. nitens* does not show a highly divergent methylome when compared to land plants. However, *K. nitens* has fewer copies of DNMT-encoding retrotransposons, which may decrease the chances for accidental retrotransposon-derived Dnmt3 activity on the genome and thus not pose a challenge to the host epigenome regulation.

The specific role of DNMTs and cytosine methylation in each of the retrotransposon families remains unclear, and likely it has host-specific and class-specific roles. However, purifying selection on the DNMT domain in all tested retrotransposons and their conservation for more than 50 million years of *Symbiodinium* evolution[24], suggests that DNMTs are likely to have a role during the retrotransposition process. For instance, LINE retrotransposons are well known to encode an RNA-binding protein as ORF1[12], thus the DNMT might be able to bind and methylate RNA. Such putative RNA methylation activity could explain the inability of the tested SymbioLINE-Dnmt to de novo methylate DNA in yeast. Unlike LINE elements, DIRS retrotransposons synthesise cDNA prior to insertion in the host genome[13]. Given the position of the DNMT domain within the *Pol* polyprotein and its ability to de novo methylate CGs, it is likely that the cDNA is methylated upon retrotranscription. As CG methylation is widespread in the *Symbiodinium* genomes, methylated CGs in retrotranscribed cDNA may be a signal to avoid being recognised as exogenous DNA. Unlike most eukaryotes besides plants and animals, *Symbiodinium* genomes encode three paralogues of methyl-binding proteins (Supplementary Data 1), which could be involved in this self-recognition mechanism. cDNA methylation could also be involved in protection against editing and degradation by mechanisms analogous to APOBECs in vertebrates[25], where deamination of cytosine into thymine would be inhibited by the addition of the methyl group, avoiding mutagenic effects. Interestingly, DIRS retrotransposons from other eukaryotic lineages encode adenine methyltransferases[13, 14]. Despite adenine and cytosine DNMTs being structurally unrelated, they could have analogous functions in transposons given the mounting

evidence of adenine methylation having regulatory roles in eukaryotic genomes[26, 27]. As DNA methylation is specifically interpreted depending on its base, sequence context and genomic location, the methylation status of the newly retrotranscribed copies of DNMT-encoding retrotransposons may affect the chromatin configuration of the insertion site. Nevertheless, given the ability of non-DNMT retrotransposons to retrotranspose in these genomes, these hypothetical defence mechanisms that the host might have to target unmethylated DNA can be circumvented by other mechanisms other than encoding a DNMT. Finally, conservation of the DNMT in the retrotransposon might be beneficial to the host, as it might slow down the retrotransposition rate by an unknown mechanism and increase the host fitness by reducing potential harmful insertions, thus secondarily increasing the retrotransposon fitness.

Many examples of domesticated proteins with transposable element origin have been described across eukaryotes[28]. These domesticated transposon-derived proteins acquire roles beneficial for the host ranging from transposon control, recruitment of cis-regulatory motifs or telomere elongation. In many cases, the process of domestication takes advantage of the close relationship between transposon machinery and the chromatin/epigenetic environment in which these elements have to navigate to survive in their hosts[28]. This link between transposable elements and epigenetic regulation makes them amenable to be co-opted for new functions by the host. But this relationship is not unidirectional. In this work, we show how transposons are also able to steal genes involved in epigenome regulation from the host. There are several reports of recurrent acquisition of protein domains by retrotransposons (RNase H, Chromo), mostly recruited from other transposable elements but also from host genes[29-31]. However, incorporation of de novo cytosine methyltransferases is a new and exciting event in the war between host genomes and their transposable parasites. Given the widely conserved roles of cytosine methylation as a transposon silencing mechanism across eukaryotes, finding active DNMTs encoded within retrotransposons in at least two independent eukaryotic lineages challenges our knowledge of retrotransposon biology and epigenome regulation in eukaryotes.

## Methods

**Algae culture and nucleic acid isolation**. *Symbiodinium kawagutii* strain CCMP2468 (Clade F1) was grown in L1 growth medium (without silicate) at 25 °C under a 14 h/10 h light/dark cycle, with a photon flux of 100 µE m$^{-2}$ s$^{-1}$. Stationary stage cultures were grown for 48 h at 25 and 32 °C, then cells were harvested by centrifugation at ~4500 × g for 5 min. Cells were first resuspended and incubated with 400 µl DNA lysis buffer (10 mM Tris pH 8.0; 100 mM EDTA pH 8.0; 0.5% SDS with 200 µg ml$^{-1}$ proteinase K) at 55 °C for 3 days. Next, cell supernatant was transferred into a new tube after centrifugation at 12,000 × g for 5 min, and the cell pellets were homogenised in lysis buffer using bead beating with 0.5 mm-zirconia/silica beads (Biospec Products, INC, Bartlesville. OK 74005) on a MP Fast Prep-24 Tissue and Cell Homogeniser (MP Biomedicals; Solon, OH 44139) at 6.0 M/S for 1 min and repeated four times. Then, the supernatant and the cell homogenate were combined, and DNA isolation was obtained using Zymo DNA extraction kit (Catalogue #D3050). RNA from the same cultures was extracted using TRIreagent (Molecular Research Center, USA), and further purified using Direct-zol RNA Miniprep (Zymo research, USA).

Cultures of *Symbiodinium* sp. (donated by Professor Roberto Iglesias-Prieto, Mexico) of clades B (ITS2 type B2) were grown in f/2 medium at 25 and 32 °C, 12:12-h day–night period, irradiance of ~50 µE m$^{-2}$ s$^{-1}$. Cultures were grown for up to 5 days, when cells were centrifuged at 12,000 × g, resuspended in the lysis buffer (100 mM Tris pH 9.0; 100 mM EDTA; 1% SDS, 100 mM NaCl with 200 µg ml$^{-1}$ proteinase K) and incubated at 65 °C for 2 h. After incubation, the samples were centrifuged at 12,000 × g (5 min) and the supernatant was transferred to a new tube. Then, the DNA was purified by phenol/chloroform/isoamyl alcohol extraction and isopropanol precipitation.

The *K. nitens* strain NIES-2285 was cultured in 50 ml BCDAT medium + 0.1% glucose by bubbling air through the medium under continuous light (10 µmol photons m–2 s –1) at 23 °C. The cells grown for 2 weeks were harvested by vacuum filtration through a nitrocellulose transfer membrane (Protran, 0.45 µm pore size, Whatman). The pelleted sample was powdered with a mortar in liquid nitrogen,

and genomic DNA was extracted using the DNeasy Plant Mini kit (Cat. No. 69104, Qiagen).

**Genome annotation**. To improve the annotation of *Symbiodinium kawagutii*, we sequenced two libraries of 100 bp single-end stranded RNA-seq (Illumina TruSeq mRNA Stranded kit) and mapped them to the published genome assembly[7] with Hisat2[32], capping the maximum intron length to 20 kb and using '—dta' option. The aligned reads were then assembled into transcripts using Stringtie with default parameters[33]. Transdecoder[34] was used to predict the coding regions of those transcripts using Pfam hits to select the best frame of each transcript. In parallel, we used the RNA-seq libraries to build a de novo transcriptome assembly using Trinity[34], which was also processed using Transdecoder. We used the resulting assemblies to characterise the actively transcribed genes in *S. kawagutii*.

In parallel, we used manually curated *Symbiodinium* retrotransposon sequences as query in a tblastn search against the genome assembly. Those hits were transformed into a hints file for Augustus ab initio gene prediction software[35], which was run in a mode to detect 'intronless' gene models guided by the hints. New transposable element candidates obtained from this approach were blasted again against the reference genome. Expanding the hit coordinates by extra 7 kb 5′ and 3′; we locally searched for LTR using RepFind[36]. To discriminate between both types of DNMT genes described in Fig. 1b, we used the Stringtie annotation to find the exon-rich genes and the intronless Augustus annotation to find the monoexonic genes.

For *S. minutum* and *K. nitens*, we used RNA-seq publicly available (SRA SAMD00008690 and SRX718727) and followed the same steps as in *S. kawagutii*. We further added the genomes of *S. microadriaticum*[37] and *S. goreaui*[38], and annotated their transposons using the same method than for *S. kawagutii*.

To quantify gene transcriptional levels, we used a reference transcriptome based on the best single isoform from the Stringtie annotation and the coding regions of transposable elements. Each RNA-seq library was then quantified using Kallisto 0.42.3[39] and Hisat2 uniquely mapped reads.

**DNMT and retrotransposon searches**. In all our searches, we used a data set of eukaryotic proteomes (Supplementary Data 1) selected to maximise the span of phylogenetic diversity. To scan the proteomes, we used Hmmer3[40] using candidate Pfam domains as query (e.g., DNA_methylase PF00145). For the resulting protein hits, we obtained the complete domain architecture using Hmmscan with PfamA database as obtained in March 2015[41]. For the dinoflagellate species without reference genome, we downloaded transcriptome assemblies from the MMETSP project[42] and we used Transdecoder to obtain coding sequences. Reference retrotransposons sequences were obtained from RepBase (download February 2014)[43]. Viral sequences were obtained from NCBI non-redundant database using retrotransposon DNMT sequences as query. *Symbiodinium* DIRS elements have a divergent reverse transcriptase that does not align well with current Pfam domain seed alignments. To increase our recovery rate of DIRS retrotransposons in *Symbiodinium*, we generated an alignment using representative sequences from divergent clades within DIRS retrotransposons that we trimmed manually and converted into an HMM profile using hmmbuild.

**Phylogenetic reconstruction**. Multiple sequence alignments were built using MAFFT[44], choosing L-INS-I mode for smaller data sets and automated mode for more than 1000 sequences. Resulting alignments were trimmed using TrimAL with '–automated1' mode[45]. For phylogenetic reconstruction, we used IQ-TREE with the in-built automated test to choose the best substitution model for each tree[46]. Branch support was computed for all trees using 100 replicates of parametric bootstrap, and 1000 replicates of the approximate likelihood ratio test and ultrafast bootstrap.

**Transposon sequence composition and selection analysis**. To characterise expansion dynamics for each class of transposable element and for each *Symbiodinium* species, we performed blastp searches all versus all and gathered the amino acid identity values for the ORF encoding the reverse transcriptase for every pair of sequences (excluding the query sequence hit against itself). For DNA transposons, we used the transposase ORF as they lack a reverse transcriptase. To establish a subset of young transposons, we used up a cutoff of 0.8 identity to establish that two transposable elements were closely related within a genome. To reveal how many active lineages there were among the young transposons, we built a network using the identity values as edges and the sequences as nodes, obtaining the total number of orthologous clusters using the 'igraph' package for R in CRAN.

To characterise the evolution of specific transposon subfamilies, we first clustered transposable element sequences belonging to all *Symbiodinium* species using blastp pairwise 0.6 identity threshold to cluster modules of conserved subfamilies across species. We then counted the numbers of each subfamily in each genome, and inferred the evolutionary dynamics using Count[47] under Dollo parsimony. DNMT and non-DNMT encoding transposons were analysed separately.

The observed versus expected ratio of CG dinucleotides on transposons and genomes data was computed as: (total number of CG dinucleotides)/((total number of Cs + total number of Gs)/2)$^2$ as in ref. [48].

To obtain the Ka/Ks ratios for the DNMT domain in retrotransposons, we gathered all retrotransposon sequences encoding DNMTs and extracted the DNMT and reverse transcriptase domain using Hmmer3[40] with custom hmm profiles. We removed identical sequences at the amino acid level and we used MAFFT[44] to align the proteins and PAL2NAL[49] to obtain the codon alignments. We filtered out sequences with too many gaps as they dramatically decreased the final size of the codon alignment. Furthermore, we only used transposable elements were both DNMT and reverse transcriptase domains passed the quality filters. Finally, we obtained the Ka, Ks and Ka/Ks values using the 'seqinr' package for R in CRAN. For interspecies comparisons, we preselected candidate sequences from all lineages depicted in Fig. 2a, b and used tblastn to obtain the best hit in each of the four *Symbiodinium* genomes. From those sequences, we selected the hit with highest bitscore not belonging to the query species. Then, we extracted the nucleotide sequences and translated them to protein using Transdecoder[34] and followed the same alignment strategy as for the intra-species comparisons. The same method was used for KnitGypsy-Dnmt3 copies belonging to the same genome.

**MethylC-seq**. To prepare the libraries pooled, the initial genomic DNA (100 ng to 1 µg) with spiked-in unmethylated lambda as control. Each sample was fragmented using a Covaris sonicator to an average size of 200 bp. This DNA was purified, end-repaired and ligated to NEXTflex methylated sequencing adaptors (BIOO Scientific). Bisulfite conversion was performed using EZ DNA Methylation-Gold Kit (Zymo Research) according to the manufacturer's instructions. The library was then amplified using KAPA HiFi HotStart Uracil + DNA polymerase (Kapa Biosystems), using seven cycles of amplification for *S. kawagutii* samples, nine for *S. minutum* and eight for *K. nitens*. Single-end 100 bp reads were sequenced for each library in an Illumina HiSeq 1500. The bisulfite-converted reads were then quality trimmed and adaptors were removed using bbduk2 included in BBMap (mink = 3 qtrim = r trimq = 10 minlength = 20). For the mapping to the reference genomes, we used BS-Seeker2[50], relying on Bowtie2 end-to-end alignments, and using Sambamba rmdup to eliminate PCR duplicates. Methylation calls were genotype corrected using read data from BS-Seeker2 ATCGmap output file. The scaffolds corresponding to plastid or mitochondrial sequences were excluded for global methylation analysis.

In each library, we spiked in an unmethylated lambda phage genome as control, which then we used to calculate the bisulfite reaction non-conversion rate (% of mC for all covered Cs) for each library.

Methylation data were visualised using IGV genome browser and DeepTools2[51]. To obtain differentially methylated regions, we used the DSS package from Bioconductor, allowing a minimum delta of 0.2 spanning at least 5 CGs, with a *p* value threshold of 0.01. Resulting DMRs were filtered to avoid regions with low coverage. All plots and statistical tests were performed using R.

**Methylation induction experiments**. We selected the target DNMTs based on domain conservation and/or expression level of the retrotransposon copy, arguing that expressed copies are more likely to be active. The DNMTs were amplified from genomic DNA of *S. kawagutii* and *K. nitens* using the proof-reading Q5 polymerase. The HA tag and the nuclear localisation signal were added in the primer sequence during PCR reaction. Amplified copies were cloned into the pYES2 vector plasmids using restriction enzymes.

We used the yeast strain BY4742 (Euroscarf, *MATα his3Δ1 leu2Δ0 lys2Δ0 ura3Δ0*), which is isogenic to S288c. After transformation with the pYES2 constructs, yeast cells were grown overnight at 30 °C in glycerol-lactate medium (0.14% YNB, 0.5% ammonium sulphate, 0.05% glucose, 2% lactate and 2% glycerol) supplemented with the required nutrients but without uracil to maintain plasmid selection. The next morning, the $OD_{600}$ of the cultures was adjusted to 0.3 $OD_{600}$ ml$^{-1}$ and cells were grown to stationary phase for 24 h after induction of the *GAL1* promoter by addition of 2% galactose.

Twenty-four hours after galactose induction, 5 ml of stationary phase cultures were mechanically lysed with glass beads. Genomic DNA was extracted using a phenol/chloroform/isoamyl alcohol extraction and was precipitated in ethanol 100%, washed with ethanol 70%, air-dried and resuspended in 80 µl TE pH 8, 10 mg ml$^{-1}$ RNAse A. DNA extractions were done for two independent replicates. MethylC-seq libraries were prepared as above, using five cycles of amplification. Mapping was performed against BY4742 genome with the pYES2 insert specific to each library appended as an extra chromosome. Methylation levels were computed as before for nuclear chromosomes, pYES2 insert and chromosome lambda.

To test for protein expression in each line, total protein extraction from the same yeast cells was performed by the NaOH-TCA lysis method. Samples were then separated on 10% SDS-PAGE gels (Invitrogen) and transferred to nitrocellulose membranes. Western blot analysis was performed using the monoclonal anti-HA (12CA5 1:1000, Roche) antibody to detect the HA-flagged inserts and the monoclonal anti-Dpm1 (5C5A7 1:2000, Invitrogen 1878306) antibody as a loading control.

**Data availability**. MethylC-seq and RNA-seq data were deposited in the Gene Expression Omnibus, GSE104474. Phylogenetic trees, hmm profiles, transposable element annotations and genome annotations and all relevant data are all available in a FigShare repository 10.6084/m9.figshare.5797812.

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

## Acknowledgements

We thank Geoff Faulkner, Marina Oliva and Ozren Bogdanovic for critical reading of this manuscript, Sophie Dove (the University of Queensland) for *Symbiodinium* sp. cultures and Oliver Rackham for pYES2 plasmids. This work has been supported by the Australian Research Council (ARC) Centre of Excellence program in Plant Energy Biology (CE140100008) and the Natural Science Foundation of China (grant #31661143029) and the National Key Research and Development Program of China grant 2016YFA0601202. R.L. was supported by an ARC Future Fellowship (FT120100862), a Sylvia and Charles Viertel Senior Medical Research Fellowship and a Howard Hughes Medical Institute International Research Scholarship. A.d.M. has been supported by a EMBO long-term fellowship (ALTF 144-2014). S.L. has in part been supported by Gordon and Betty Moore Foundation (GBMF grant #4980). P.L. was supported by intramural funding from CNRS, the Université Paris Diderot and the Institut National de la Santé et de la Recherche Médicale, A.B. has been supported by a post-doctoral fellowship from the ANR through the initiatives d'excellence (Idex ANR-11-IDEX-0005-02) and the Labex 'Who am I?' (ANR11-LABX-0071). All data sets have been deposited in GEO under the accession number GSE104474.

## Author contributions

A.d.M. and R.L. designed the study. A.d.M. prepared MethylC-seq and RNA-seq libraries, which were sequenced with the help of J.P. The data were analysed by A.d.M., with help from S.B. Cloning was performed by A.d.M. and D.B.V.-L. Yeast work was performed by A.B. and P.L. *Symbiodinium* materials and experiments were provided by S.L., N.R., N.J., F.H., F.Y. and L.L. *Klebsormidium* materials were provided by H.O. and K.H. The manuscript was written by A.d.M. and R.L.

## Additional information

**Competing interests:** The authors declare no competing interests.

