## [Peer Review File · Nature Communications]

Reviewers' comments:

Reviewer #1 (Remarks to the Author):

The manuscript "Convergent acquisition of cytosine methyltransferases into eukaryotic retrotransposons" describes the analysis of at least two independent expansions of DNA Methyltransferase domain containing genes in the dinoflagellates and a charophyte lineages. A careful annotation and phylogenetic analysis of these expansions finds that they are almost certainly the result of acquisition by a retroelements, and subsequent amplification during transposition. In the case of the dinoflagellates (specifically Symbiodinium) it appears that this process has occurred multiple times involving different DNA methylases and retroelements. The authors confirm the activity of some of the methylases in the heterologous yeast system, and go on to characterize the DNA methylation and transcriptional landscape of these species eventually arguing that changes in genome wide DNA methylation patterns and the recruitment of DNA methyltransferases by retroelements represents ongoing adaptive coevolution between retroelements and their host genomes.

The manuscript is very clearly written and understandable. Overall, I found the central finding of the study, the recruitment of CMTs by transposons, really exciting. While acquisition of protein coding genes by transposons has been described previously, the finding that transposons have acquired a potentially important factor in their own regulation is very intriguing. It is also interesting that there seems to be a change in DNA methylation patterns in one of the analyzed lineages. While the authors make a valiant effort to connect these two things, their observations remain correlative, and not wholly consistent. As a result, I think that many of the authors claims are overstated, and in general I think that manuscript would be better served by explicitly stating weaknesses / questions that remain open. My specific comments follow:

The authors propose recruitment of CMT by retrotransposons to prevent targeting (via an unknown mechanism). To my knowledge, it is not shown that host DNA is discriminated from non-host cDNA by DNA methylation in any eukaryotic system. While this hypothesis is intriguing, I think that the manuscript only hints at this possibility. It makes sense to mention this idea once in the discussion, but to include it in other interpretations (and in the abstract) is a stretch.

The authors do not discriminate between the evolutionary age of the CMT retro lineage, and the age of the observed expansions. Thus it is still possible that the observed CMT retro result from several recent expansions of relatively few elements.

In line with the above authors should compare the retroelement content of these genomes to other genomes. Is elevated activity of LINE elements in these genome in general a characteristic of this species? Might this make rare elements such as this more likely?

The connection between GC%, genome-wide methylation patterns, and the recruitment of CMT domains to retroelements has an N=1. This alone would be a tenuous basis to put forth the hypothesis of a relationship made here, but the authors do not find that this relationship is evolutionarily replicated. While not all important coevolved traits will necessarily be replicated during evolution, the functional hypothesis relating the retroelement and genome-wide patterns is completely correlative in this case and should be more overtly acknowledged throughout the manuscript.

The authors do not show that DNA methylation plays a role in transposon silencing in this lineage (or related lineages), nor do they show that the newly discovered CMTs have any role in promotion or repression of transposition. Statements to this effect should be softened or removed.

Line 163-175: While cytosine deamination has been implicated as a potential factor in changes in genome GC%, it is by no means the only factor or even the major factor. Biased mismatch repair, genomic locality, and recombination rates are all important contributors to GC bias, and not addressed here. Furthermore, the GC bias in substitutions is well known to be dependent on substitution age. Is the average evolutionary age of each of these element classes comparable? I suggest the authors either remove this paragraph/analysis (I don't really think it is necessary to support their point) or should expand the analysis to explore other possible explanations. The transposon based evolutionary trade off suggest by the last sentence of this paragraph is interesting, but neglects the host's fitness (on which the retrotransposon is dependent). It is entirely possible that the most "fit" transposons from the standpoint of replication are never seen in extant genomes because they are deleterious to the host.

Line 179: No explicit test is done and the presentation method in figure 2e only allows outliers to be shown. In other words, the analysis does not explicitly test whether "transposable element families bearing a CMT are more likely to transcribed...". Furthermore, the authors find varied patterns in amongst different elements. So, the answer to the stated question for this analysis is "There is not clear relationship between expression and the presence of a CMT". Instead the authors claim that their data highlight "the ability of these retrotransposons to escape from the host silencing mechanisms". I do not think that the authors have demonstrated that this makes them different from other retroelements.

Suggestions and minor comments:

Line 30: Remove surprisingly

Line 38: Remove unprecedented

Line 38: This sentence claims something not shown by the paper. No in vivo role for these CMTs has been demonstrated

Line 69-70: The criteria for delimiting the two categories of genes is not clearly laid out here or in the methods. I assume it is simply 1) genes with introns and 2) genes without introns, but this should be explicit. Also, why are numbers from just one species listed here?

Line 68: I suggest that "abundance" be changed to something like copy number to discriminate from gene expression.

Line 76-88 and Figure 1D: Were any exon rich copies of Dnmt3 found? Why does the figure not include DIRS elements? A full table listing exon rich + intronless numbers for each class (rather than shaded table) would be more informative.

Line 218-218: Needs citation

Line 219-221: Why do the authors claim a 10% change between species and a 5% global change in response to treatment as "similar"? These are enormous changes given the size of the dataset.

Sup Figure 7: Why are the CG and CH thresholds different here? At minimum CG data at the same threshold should be shown. Also, the data presented kawaguti watson/crick strans appears near categorical suggesting low coverage. The datasets should be assessed at similar levels of coverage based on downsampling to compare correlation coefficients.

Line 238-239: What is the spearman rank for other systems?

Line 243-246: What statistical power do the authors have with this dataset? They are basing a claim on a negative result, so the authors must demonstrate that they should identify DMRs given this design.

Line 262: "...and likely contributes..." This is speculation and should not be included in the results.

Line 313: Why is no correlation value reported here?

Line 318-323: This hypothesis belongs in the discussion. In addition, since the null hypothesis is that there is not a universal relationship between CMT recruitment and genome wide methylation patterns, it is fine that they do not observe it here.

Line 362-366: The authors do not provide a positive control for CH methylation and thus can not make

conclusions regarding the activity of these proteins as transposon-like or gene-like.

Line 418: Needs reference

Line 425: Should be made clear that this is a contribution from this work

Line 425-427: I agree that this is potentially exciting, though I think this manuscript only presents a hypothesis rather than shows this is what happens.

Line 428-9: This statement is far too strong. The authors need to present functional evidence that "hijacking" has occurred.

Reviewer #2 (Remarks to the Author):

This paper starts from the observation that the genome of the dinoflagellate *Symbiodinium* is replete with intronless DNA methyltransferase domains that have been captured by retrotransposons. The peculiarities of the *Symbiodinium* methylome are fascinating, although suggestions as to how this is related to conflict with retrotransposon-DNA methyltransferases remain somewhat speculative. The results presented here suggest that the field has not yet exhausted the insights that can be garnered from genome and methylome analysis, both about the evolutionary dynamics of DNA methylation and its function, especially with regards to less well-studied species like dinoflagellates. I'm generally supportive of publication after the comments below are addressed.

Specific comments:

The use of the abbreviation CMT for cytosine methyltransferases is confusing because the chromomethyltransferases are abbreviated as CMTs in land plants. It is recommended that the authors use the more commonly accepted abbreviations DMTs or Dnmt for DNA methyltransferases.

In the abstract, line 35: "most of the genome is hypermethylated at CG dinucleotides". Hypermethylated compared to what? Saying "methylated at CG dinucleotides" is sufficient.

Line 94 and throughout should be "reverse transcriptase" not "retrotranscriptase"

Line 163-164: more accurately, the recurrent acquisition of CMTs suggests an advantageous role of the cytosine methyltransferase domain, not of cytosine methylation.

Line 167: how was the expected ratio of CGs in retrotransposons calculated?

Line 173: "SymbioLINE-Dnmts carry CMTs that impose such a mutational burden upon themselves...." This statement seems too speculative. The methylation of these retrotransposons could come from the captured CMT or from the conventional DNA methyltransferase genes elsewhere in the genome.

Lines 231-234: another possibility is that the methylation variation is from individual to individual (or sexual phase vs vegetative phase?), not cell to cell.

On line 321 the authors conclude that CG methylation is mostly not related to silencing in *K. nitens*. How do they know that? Yes, CG-only methylation in gene bodies is not correlated with silencing, as in angiosperms, but the pattern of TE methylation alone, with both CG and non-CG methylation present, does not give insight into functionality here.

In the yeast experiments, are the newly methylated CGs symmetric, or is methylation asymmetric as in endogenous *S. kawagutii*?

Lines 383-390: The logic as to why ubiquitous CG methylation is a response to methylation by the retransposon encoded DNA methyltransferase, or why basic functions of cytosine methylation have been lost, is not clear to me. The authors should expand on their thinking here.

Reviewer #3 (Remarks to the Author):

In this manuscript, authors describe the acquisition of cytosine methyltransferases by eukaryotic retrotransposable elements in two lineages: the dinoflagellate, *Symbiodinium kawagutii*, and a filamentous alga, *Klebsormidium nitens*. The detailed description of these kind of occurrences and their functional consequences for the retrotransposable element fitness and the genome would be very interesting, but I do not think the work gets there.

Authors should provide more details and analyses of the sequences to back up their claims and be cautious not to over interpret their results. It is OK to answer only a fraction of the questions and leave others unanswered. For example, I do not think the authors have shown yet that these acquisitions have the same effect in the different species or TEs, i.e., these acquisitions are an example of convergence, or are adaptations of the transposable elements (TEs) to the host epigenomic environment. I also think that authors should facilitate the access to all the genomic information.

Detailed comments:

1. To facilitate access to the genomic information (TE copies and TE copies with CMT acquisitions) to readers/users not familiar with those genomes, links to the genome browsers for *Symbiodinium kawagutii* and *Klebsormidium nitens*, sequences and annotations of those sequences (TE copies and TE copies with CMT acquisitions) and a quick characterization of their distribution in the genome should be provided. Authors need to make easy for a reader to find those examples and understand their impact in the genome.
2. Authors should also mention why they think that these TE copies with CMT acquisitions are not artifacts of the genome assemblies and provide details of any confirmatory results they have.
3. I do not think the authors have shown yet that these acquisitions are "adaptations of the transposable elements to the host epigenomic environment" as claimed but it would be very interesting, if they were. At this point I only see one way to get that data. Authors could study the expansion of the different TE lineages in *Symbiodinium kawagutii* (i.e., the several SymbioDIRS structures with and without CMTs) and show what lineages have been more successful than others in the recent past (i.e., have produced more copies in the recent past). This would enhance the publication a lot.

Conservation of the CMTs does not per se prove that they are beneficial for the TE transposable elements. Since no copy with introns of Dnmt3 is found, it could be that those are the source of Dnmt3 now (See point 5).

Authors argue that "the fact that SymbioLINE-Dnmts carry CMTs that impose such a mutational burden upon themselves should only be tolerated if they confer beneficial effects to the retrotransposon." The reality is that the retrocopy insertions of CMTs on TEs could make the fitness of the TE lower and we could still see copies as long as the fitness is not zero. So, a study of the recent success of the different TE lineages is the only way I see to test the TE fitness.

4. The Introduction about cytosine methyltransferases should be more general because different kinds of methylation have different effects and it is also genome dependent (e.g., plants vs. animals) and authors discuss all of this later including a potential role related to RNA editing (Discussion). This also relates to the alleged convergence. Authors observe CMTs in LTR and Non-LTR elements and they likely have different functional consequences for the different classes of TEs that have very

different retrotransposition mechanisms and might not be examples of convergence.

5. The origin of the acquired CMTs is unclear (e.g., Dnmt3 of SymbioDIRS-Dnmt3). Where is the "parental" gene, i.e., gene with introns? Has it been lost? Is it a pseudogene or is it the portion of the genome that has not been sequenced? Have the retrocopies replaced that parental gene and this is why they are under purifying selection but they are not really beneficial for the TE? Or does the CMT come from a different species?

6. Page 7. It is interesting that authors observe complete and not disabled copies of the TEs with acquired CMTs in different lineages. I am guessing they are comparing the consensus sequences but they should comment on this. It would be interesting to test if K_a/K_s between species for the same TE family that was acquired before the species split is significantly smaller than 1 although as mentioned in point 5 that might not be enough to show if these arrangements are beneficial for the TE but it would support that they have stayed functional even when the functional analyses in yeast are not conclusive.

7. Page 12. There are aspects of the functional analyses of methylation that are not conclusive and authors should be clear about it. For example, there is a lack of correlation between methylation. TEs appear not to be silenced in all organisms (i.e., not in all the cells). Authors propose methylation occurs upon insertion but how does it end being so dynamic?

8. In several places in the text and figures authors assign expression to particular genes or TE copies (e.g., Page 3 or Page 8-Supplementary Figure 6). How do authors know what RNA-seq reads belong to the particular gene or copy depicted? Explain.

9. Page 19 – It is not clear what authors mean by "ubiquitous CG methylation in the Symbiodinium genome could be a response of the host to uncontrolled retrotransposon CMT methylation". They should explain better this hypothesis.

10. Page 21. Authors state "Many examples of domesticated transposable elements have been described across eukaryotes. These domesticated transposons acquire roles beneficial for the host ranging from transposon control, recruitment of cis-regulatory motifs or telomere elongation." This should be rewritten. As it is currently written, it appears that the whole TE is domesticated but I think the authors mean TE protein domestication.

Reviewer #4 (Remarks to the Author):

The authors reported retrotransposons encoding cytosine methyltransferases (CMTs). These transposons were found in two eukaryotic lineages, dinoflagellates and charophytes. The authors also show that the genome of Symbiodinium have high level of cytosine methylation at CpG sites. Finally they show some of the transposon-encoded CMTs have activities to methylate CpG sites in the yeast.

In my opinion, it remains unclear how the transposon-encoded CMTs contribute for success of the transposons. The authors speculated that retrotransposons could self-methylate retro-transcribed DNA as a way to mimic host genomic DNA, but the evidence supporting that is limited. Without further evidences, their findings are rather descriptive. Overall, I feel the paper is too preliminary for publication in Nature Communications. Below are specific questions and comments, which might be useful for them.

1) Do unmethylated transposon copies silenced or excluded from the genomes of these species more efficiently than methylated copies? The authors should show that for supporting their speculation, because cytosine methylation function in the opposite way in vertebrates and plants; in these organisms, transposons are silenced by cytosine methylation.

2) When the full-length transposons are introduced into yeast or other species without genomic DNA methylation, do the CMTs self-methylate the transposons more efficiently than the other regions

within the genome? Normally CMT can function in trans, and I do not understand why each of the transposons should keep the CMT genes.

3) An alternative possibility for the presence of CMTs in the transposons could be silencing of endogenous CMTs by RNAi? If so, host CMTs evolve rapidly to escape from the silencing by the transposon-encoded CMTs and the transposons incorporate the new CMTs constantly from the host genome to silence them. Do the transposon-encoded CMTs evolve rapidly in the nucleotide sequence level? Are small RNAs for CMTs detected in these species?

Reviewer #5 (Remarks to the Author):

The manuscript by de Mendosa et al. reports a thorough analysis of retrotransposon-associated methyltransferase domains in poorly studied taxa, dinoflagellates and charophytes. It is well-written, technically solid, and provides a comprehensive inventory of MTases, as well as the data on transcription, bisulfite sequencing, and activity of selected MTases in a heterologous system.

While the subject is certainly of interest, the MS nevertheless leaves the reader unsure what the data actually means, and whether it should be interpreted in the way it is presented in the abstract. To promote the presented concept that retrotransposons acquire MT genes repeatedly to mimic themselves as host genomic DNA, the authors need to demonstrate the existence of host response to unmodified genomic invaders in this particular host. Barring that, the null hypothesis could be that the transposon-encoded MT may be just decorating DNA with little or no consequences for either the element or the host. Could some MTs, including the ones from obscure clades, be circulating in the aquatic environment, e.g. in viruses, and be repeatedly captured from such sources by horizontal means, and eventually lost? It would be good to see convincing evidence that they were indeed captured from the host, as the MS claims. In Fig. 1, this information is not made visible, i.e. where the intron-containing MTs are placed on the tree - only the domain structure is presented in Fig. S1. Are RT and MT domain topologies fully congruent? Maybe they can share/exchange components?

Also, if the profiles for CG and CH differ, different enzymes would likely be responsible for modification - since DIRS MTs favor CG, perhaps one of the host MTases is at work?

Specific comments:

-No analysis of RT-MT retrotransposons in the published *Symbiodinium microadriaticum* genome (Sci Rep 2016, 6:39734) is presented. It is reported to be ancestral to *S. minutum* and *S. kawagutii*, and to contain ~2700 copies of DIRS elements, some of which are MT-containing. Does it show similar CG under-representation in those transposons?

-The presence of MT in DIRS elements has been known for a long time, and the list of references to MT-encoding DIRS could be significantly expanded (e.g. fungi, nematodes): Muszevska et al. PLOS One 2013, 8(9); Lunt and coworkers PLOS One 2014, 9(9).

-Convergent evolution of other components, e.g. RNase H repeatedly captured by different retrotransposons (L1, gypsy), in many of them several times, was reported in diverse taxa, such as oomycetes, plants, and fungi (PNAS 2013, 110:20140; Mob DNA, 2017, 8:4; Mol Biol Evol 2015, 32:1197).

Below we provide a point by point response to the Reviewers comments and suggestions.

Reviewer #1 (Remarks to the Author):

The manuscript "Convergent acquisition of cytosine methyltransferases into eukaryotic retrotransposons" describes the analysis of at least two independent expansions of DNA Methyltransferase domain containing genes in the dinoflagellates and a charophyte lineages. A careful annotation and phylogenetic analysis of these expansions finds that they are almost certainly the result of acquisition by a retroelements, and subsequent amplification during transposition. In the case of the dinoflagellates (specifically Symbiodinium) it appears that this process has occurred multiple times involving different DNA methylases and retroelements. The authors confirm the activity of some of the methylases in the heterologous yeast system, and go on to characterize the DNA methylation and transcriptional landscape of these species eventually arguing that changes in genome wide DNA methylation patterns and the recruitment of DNA methyltransferases by retroelements represents ongoing adaptive coevolution between retroelements and their host genomes.

The manuscript is very clearly written and understandable. Overall, I found the central finding of the study, the recruitment of CMTs by transposons, really exciting. While acquisition of protein coding genes by transposons has been described previously, the finding that transposons have acquired a potentially important factor in their own regulation is very intriguing. It is also interesting that there seems to be a change in DNA methylation patterns in one of the analyzed lineages. While the authors make a valiant effort to connect these two things, their observations remain correlative, and not wholly consistent. As a result, I think that many of the authors claims are overstated, and in general I think that manuscript would be better served by explicitly stating weaknesses / questions that remain open. My specific comments follow:

Comment: The authors propose recruitment of CMT by retrotransposons to prevent targeting (via an unknown mechanism). To my knowledge, it is not shown that host DNA is discriminated from non-host cDNA by DNA methylation in any eukaryotic system. While this hypothesis is intriguing, I think that the manuscript only hints at this possibility. It makes sense to mention this idea once in the discussion, but to include it in other interpretations (and in the abstract) is a stretch.

Response: We have removed this from the abstract and we have left it just as an hypothesis in the discussion, among other possible alternatives. Moreover, we have rewritten abstract, introduction, results and discussion to explicitly state which points are still open for interpretation.

Comment: The authors do not discriminate between the evolutionary age of the CMT retros lineage, and the age of the observed expansions. Thus it is still possible that the observed CMT retros result from several recent expansions of relatively few elements.

Response: To tackle this we have now included the genomes of two new *Symbiodinium* species in the retrotransposon analysis. The *Symbiodinium* genus is very old, estimated to have its origins 50 million years ago (<https://www.ncbi.nlm.nih.gov/pubmed/15978847>). Moreover, *Symbiodinium* genomes have evolved quite fast, thus they are highly divergent at the nucleotide level. This is just to highlight that *Symbiodinium* species are not like species belonging to the same genus in plants or metazoans, but rather are quite an ancient and divergent group of protists.

Adding two more species, including one from the earliest diverging group of *Symbiodinium* (*S. microadriaticum*), we observe that all 3 types of *Symbiodinium* DNMT encoding retrotransposons were already present in the last common ancestor of the group. Moreover, when looking at the new phylogenetic trees presented in Figure 2a and 2b, we still see that all lineages have sequences in more than one clade. Together, this provides a strong indication that these retrotransposons are not recent expansions, but rather are quite ancient groups.

Finally, we have used amino acid pairwise identity between retrotransposons to establish the expansion dynamics of all TEs in *Symbiodinium* genomes (Supplementary Figure 5c), and we observe similar patterns in all families, suggesting a mix of new and old expansions.

For *K. nitens* retrotransposons, both the very low Ka/Ks ratios (Supplementary figure 12) and the branch lengths observed in Figure 4a also indicate that those are quite divergent copies, rather than very similar recently evolved copies.

Comment: In line with the above authors should compare the retroelement content of these genomes to other genomes. Is elevated activity of LINE elements in these genome in general a characteristic of this species? Might this make rare elements such as this more likely?

Response: Mammals and vertebrates show higher numbers of LINE elements than *Symbiodinium* (<https://www.ncbi.nlm.nih.gov/pubmed/27702814>), yet they have not evolved such forms. We do not think this is simply a byproduct of high LINE activity that by random chance has acquired a DNMT domain that is passively maintained. If that was the case, we would possibly observe just a recent expansion of SymbioLINE-Dnmt. Rather, they date back at least 50 million years, they are highly divergent at the nucleotide/amino acid level across copies, several lineages seem to be retro-transcriptionally active, and we have further shown that DNMTs are under purifying selection (Supplementary Figure 8). Therefore, we agree that having many active LINE elements necessarily had a role in the early acquisition of the DNMT, as it increased the chances of placing a LINE element next to a DNMT gene. But further expansion of the lineage and conservation of the DNMT domain strongly supports it having a function. We now mention this comparison to other lineages and its implications in the results section.

Comment: The connection between GC%, genome-wide methylation patterns, and the recruitment of CMT domains to retroelements has an N=1. This alone would be a tenuous basis to put forth the hypothesis of a relationship made here, but the authors do not find that

this relationship is evolutionarily replicated. While not all important coevolved traits will necessarily replicated during evolution, the functional hypothesis relating the retroelement and genome-wide patterns is completely correlative in this case a should be more overtly acknowledged throughout the manuscript.

Response: We do now acknowledge that this connection is just a possibility in the revised discussion and abstract. It is true that it cannot be ruled out that the retrotransposon encoded DNMTs have no effect in the epigenome composition, as we observe in *K. nitens*.

Comment: The authors do not show that DNA methylation plays a role in transposon silencing in this lineage (or related lineages), nor do they show that the newly discovered CMTs have any role in promotion or repression of transposition. Statements to this effect should be softened or removed.

Response: We have revised the manuscript to avoid making these claims.

Comment: Line 163-175: While cytosine deamination has been implicated as a potential factor in changes in genome GC%, it is by no means the only factor or even the major factor. Biased mismatch repair, genomic locality, and recombination rates are all important contributors to GC bias, and not addressed here. Furthermore, the GC bias in substitutions is well known to be dependent on substitution age. Is the average evolutionary age of each of these element classes comparable? I suggest the authors either remove this paragraph/analysis (I don't really think it is necessary to support their point) or should expand the analysis to explore other possible explanations. The transposon based evolutionary trade off suggest by the last sentence of this paragraph is interesting, but neglects the host's fitness (on which the retrotransposon is dependent). It is entirely possible that the most "fit" transposons from the standpoint of replication are never seen in extant genomes because they are deleterious to the host.

Response: First we apologize if it was not clear in the previous version of the manuscript, but we were not referring to GC%, but rather CpG dinucleotide composition. CpG dinucleotides are just one of the combinations of dinucleotides you could obtain in any given sequence (CT, CA, GA, GG, ...), irrespectively to the GC%. CpGs are clearly depleted in many genomes, which correlates well with methylated genomes in the CG context such as vertebrates. Moreover, CpG dinucleotides mutate much faster than other dinucleotides in mammals (<http://genome.cshlp.org/content/early/2010/05/11/gr.103283.109>). However, there are species with relatively high levels of methylation and an enrichment of CpG dinucleotides (see <https://www.ncbi.nlm.nih.gov/pubmed/24630728>, or the blue bars in Supplementary Figure 6a). Therefore, we think it is quite relevant to report that *Symbiodinium* species, having highly methylated genomes in the CpG context, do show some depletion compared to the expected number of CpG you should get by random chance given the GC% content.

The same applies for retrotransposon sequences. We show that most retrotransposon types in *Symbiodinium* have similar ratios of CpG observed/expected compared to the overall genomic level. Moreover, DIRS with and without DNMT do not show significantly different ratios. However, LINE-Dnmt are significantly depleted. In the revised manuscript, we have now included the four *Symbiodinium* species in this analysis (Figure 2d) and all show the same pattern. This pattern holds if we only take "young" copies, although it is not significant

in a wilcoxon rank-sum test for two of the species, due to low numbers in *S. minutum* and *S. microadriaticum*. Therefore we think it is an interesting pattern to report.

Furthermore, we now have listed other possible causes of this depletion in the text, we are explicit about referring to CG dinucleotides and not GC%, and we have removed the trade-off sentence of the text as per the reviewer's suggestion.

Comment: Line 179: No explicit test is done and the presentation method in figure 2e only allows outliers to be shown. In other words, the analysis does not explicitly test whether "transposable element families bearing a CMT are more likely to transcribed...". Furthermore, the authors find varied patterns in amongst different elements. So, the answer to the stated question for this analysis is "There is not clear relationship between expression and the presence of a CMT". Instead the authors claim that their data highlight "the ability of these retrotransposons to escape from the host silencing mechanisms". I do not think that the authors have demonstrated that this makes them different from other retroelements.

Response: We have moved this figure to the supplementary material and we have removed the interpretations pointed out by the reviewer.

Suggestions and minor comments:

Comment: Line 30: Remove surprisingly

Response: Done.

Comment: Line 38: Remove unprecedented

Response: Done.

Comment: Line 38: This sentence claims something not shown by the paper. No in vivo role for these CMTs has been demonstrated

Response: This has been rewritten accordingly.

Comment: Line 69-70: The criteria for delimiting the two categories of genes is not clearly laid out here or in the methods. I assume it is simply 1) genes with introns and 2) genes without introns, but this should be explicit. Also, why are numbers from just one species listed here?

Response: The criteria have now been clearly stated in the *Genome Annotation* section of the Online Methods, and the missing values in the text have been fixed.

Comment: Line 68: I suggest that "abundance" be changed to something like copy number to discriminate from gene expression.

Response: Rewritten as suggested.

Comment: Line 76-88 and Figure 1D: Were any exon rich copies of Dnmt3 found? Why does the figure not include DIRS elements? A full table listing exon rich + intronless numbers for each class (rather than shaded table) would be more informative.

Response: No copies of exon-rich Dnmt3 were found. Figure 1D now specifies that all the DNMT3 genes belong to SymbioDIRS-Dnmt3.

Comment: Line 218-218: Needs citation

Response: Citation has been added.

Comment: Line 219-221: Why do the authors claim a 10% change between species and a 5% global change in response to treatment as “similar”? These are enormous changes given the size of the dataset.

Response: This has been rewritten to highlight this difference between species. Regarding the 3.7% difference in the treatment sample of *S. minutum* being considered “similar” to the control, this is because the treatment methylome has very low mappability and therefore was not sequenced deep, thus being a lower accuracy measurement.

Comment: Sup Figure 7: Why are the CG and CH thresholds different here? At minimum CG data at the same threshold should be shown. Also, the data presented kawaguti watson/crick strans appears near categorical suggesting low coverage. The datasets should be assessed at similar levels of coverage based on downsampling to compare correlation coefficients.

Response: CG and CH thresholds are the same for required coverage in the given window. However, for CG windows we required ≥ 2 CpGs because most windows lack CpG dinucleotides (the genome is depleted of CpGs, as discussed above). Instead, Cs in the CpH context are very common, thus all tested windows have >1 C (or G). For the Watson/Crick strand plot, we have now downsampled the datasets to the same coverage levels (which capture the majority of sites in all three species) and the results remain the same (high correlation in plants and mammals and very low correlation in *Symbiodinium*). Moreover, Arabidopsis has fewer sites than *S. kawagutii*, as its genome is 4 times smaller, however, its values are highly correlated. This is consistent with the bimodal distribution of methylation levels at CG sites in Arabidopsis, with sharp peaks at 0 and 1, while *S. kawagutii* shows a broad unimodal peak at 0.8 as shown in Figure 3b.

Comment: Line 238-239: What is the spearman rank for other systems?

Response: We have included the spearman rank for *Klebsormidium nitens* in the text. We chose this statistic to reflect the lack of correlation, because if we plot the methylation levels on gene bodies divided per deciles of expression as we did in Supplementary Figure 13d for *K. nitens*, we obtain are 4 overlapping flat lines, which is not a very informative figure.

Comment: Line 243-246: What statistical power do the authors have with this dataset? They are basing a claim on a negative result, so the authors must demonstrate that they should identify DMRs given this design.

Response: This algorithm is specifically tailored to detect DMRs in this type of dataset (<https://www.ncbi.nlm.nih.gov/pmc/articles/PMC4666378/>, benchmarked in <https://www.ncbi.nlm.nih.gov/pubmed/28334228>), and we have successfully used this approach in DMR identification in mammals, metazoans and simulated datasets in which DMRs are known to exist (<https://www.biorxiv.org/content/early/2017/12/02/228221>). In this case the lack of success in finding DMRs is not surprising, given the lack of unmethylated regions (which are the prime candidates for having differential methylation) in the genome of this species and the noisy nature of its methylation patterns.

Comment: Line 262: "...and likely contributes..." This is speculation and should not be included in the results.

Response: This sentence has been removed.

Comment: Line 313: Why is no correlation value reported here?

Response: The correlation value is now reported "(spearman $r = 0.427$)".

Comment: Line 318-323: This hypothesis belongs in the discussion. In addition, since the null hypothesis is that there is not a universal relationship between CMT recruitment and genome wide methylation patterns, it is fine that they do not observe it here.

Response: This sentence has been deleted.

Comment: Line 362-366: The authors do not provide a positive control for CH methylation and thus can not make conclusions regarding the activity of these proteins as transposon-like or gene-like.

Response: We have rewritten the text to avoid making claims of specificity of CG methylation. However, we would like to point out that given the lack of endogenous DNA methylation in yeast, CH and CG should be equally likely to be deposited given that there are no physical impediments or possible toxicity of one context or the other.

Comment: Line 418: Needs reference

Response: Reference added.

Comment: Line 425: Should be made clear that this is a contribution from this work

Response: Our contribution is now highlighted.

Comment: Line 425-427: I agree that this is potentially exciting, though I think this manuscript only presents a hypothesis rather than shows this is what happens.

Response: As previously, we have rewritten this part to clarify that this is a hypothesis and present our results in a more cautious manner.

Comment: Line 428-9: This statement is far too strong. The authors need to present functional evidence that “hijacking” has occurred.

Response: This sentence has been rewritten.

Reviewer #2 (Remarks to the Author):

This paper starts from the observation that the genome of the dinoflagellate *Symbiodinium* is replete with intronless DNA methyltransferase domains that have been captured by retrotransposons. The peculiarities of the *Symbiodinium* methylome are fascinating, although suggestions as to how this is related to conflict with retrotransposon-DNA methyltransferases remain somewhat speculative. The results presented here suggest that the field has not yet exhausted the insights that can be garnered from genome and methylome analysis, both about the evolutionary dynamics of DNA methylation and its function, especially with regards to less well-studied species like dinoflagellates. I'm generally supportive of publication after the comments below are addressed.

Specific comments:

Comment: 1. The use of the abbreviation CMT for cytosine methyltransferases is confusing because the chromomethyltransferases are abbreviated as CMTs in land plants. It is recommended that the authors use the more commonly accepted abbreviations DMTs or Dnmt for DNA methyltransferases.

Response: We have changed CMT for DNMT throughout the text and figures. The rationale behind previously using CMT was because the use of DNMT has a couple of potential problems. First, it assumes the substrate is DNA, which is not always the case (see DNMT2). Second, adenine methyltransferases are also DNMT, but unrelated structurally and phylogenetically to cytosine methyltransferases. We have rewritten the introduction to make this point clear for the readers, however we agree that the field tends to use DNMT to refer to this gene family and the CMT nomenclature could be confounded with plant CMTs DNA methyltransferase gene family.

Comment: 2. In the abstract, line 35: “most of the genome is hypermethylated at CG dinucleotides”. Hypermethylated compared to what? Saying “methylated at CG dinucleotides” is sufficient.

Response: This has been revised as suggested.

Comment: 3. Line 94 and throughout should be “reverse transcriptase” not “retrotranscriptase”

Response: This has been revised as suggested.

Comment: 4. Line 163-164: more accurately, the recurrent acquisition of CMTs suggests an advantageous role of the cytosine methyltransferase domain, not of cytosine methylation.

Response: The sentence has been rewritten to capture the reviewer's suggestion.

Comment: 5. Line 167: how was the expected ratio of CGs in retrotransposons calculated?

Response: We have used the formula " $((\text{total number of Cs} + \text{total number of Gs})/2)^2$ " as in <https://www.ncbi.nlm.nih.gov/pmc/articles/PMC1345710/>. Other approaches such as: " $(\text{total number of Cs} * \text{total number of Gs})/\text{sequence length}$ " produced equivalent results.

Comment: 6. Line 173: "SymbioLINE-Dnmts carry CMTs that impose such a mutational burden upon themselves..." This statement seems too speculative. The methylation of these retrotransposons could come from the captured CMT or from the conventional DNA methyltransferase genes elsewhere in the genome.

Response: This sentence has been deleted.

Comment: 7. Lines 231-234: another possibility is that the methylation variation is from individual to individual (or sexual phase vs vegetative phase?), not cell to cell.

Response: We have added the following sentence to highlight this possible source heterogeneity as the reviewer suggests: "Cell heterogeneity could reflect different replication timepoints or hidden life cycle phases amongst cells in culture". However, *Symbiodinium* cultures are quite homogeneous and the methylation patterns that we observe are not consistent with two distinct stages, but rather with a noisy deposition of mCG.

Comment: 8. On line 321 the authors conclude that CG methylation is mostly not related to silencing in *K. nitens*. How do they know that? Yes, CG-only methylation in gene bodies is not correlated with silencing, as in angiosperms, but the pattern of TE methylation alone, with both CG and non-CG methylation present, does not give insight into functionality here.

Response: The following change has been implemented in the revised manuscript: "However, in both cases, CH methylation appears to be involved in transposable element targeting, while CG methylation is targeting active gene bodies and silent transposons alike."

Comment: 9. In the yeast experiments, are the newly methylated CGs symmetric, or is methylation asymmetric as in endogenous *S. kawagutii*?

Response: We have now included this analysis in Supplementary Figure 14e. Both SymbioDIRS-Dnmt3 and human DNMT3a show asymmetric methylation (on sites where at least one of the strands shows $mC/C > 0$ and coverage is $>10x$). This is now described in the text. The lack of symmetric methylation with DNMT3A is consistent with previous reports (<https://www.ncbi.nlm.nih.gov/pmc/articles/PMC1084342/>). However, we do not think that SymbioDIRS-Dnmt3 is responsible for maintaining the genome methylation patterns of the host genome, as *S. kawagutii* has 11 Dnmt5 genes and 4 Dnmt6 genes expressed at higher levels than any DNMT encoded within a retrotransposon. However, possible bursts of

retrotransposon DNMT activity could be detrimental for a species that requires some regions to be methylated and some regions unmethylated (e.g. promoters in vertebrates and plants), which could have been a cause of the evolution of such a divergent methylome.

Comment: 10. Lines 383-390: The logic as to why ubiquitous CG methylation is a response to methylation by the retrotransposon encoded DNA methyltransferase, or why basic functions of cytosine methylation have been lost, is not clear to me. The authors should expand on their thinking here.

Response: In the revised manuscript we have expanded the discussion of this idea to improve clarity for the reader. It now reads: “In the striking case of *Symbiodinium*, given the demonstrated ability of SymbioDIRS-Dnmt3 DNMTs to methylate CGs and the pervasive transcription of these retrotransposons, it is reasonable to speculate that ubiquitous CG methylation in the *Symbiodinium* genome could be a response of the host to uncontrolled retrotransposon DNMT methylation. If locally restricted regions of the genome would require lack of methylation for proper regulation, such as promoters in vertebrates, undesired retrotransposon derived methylation could have detrimental effects on the host fitness. This could explain why some basic functions of cytosine methylation have been lost in this lineage through methylation of all CGs as the default state, which would represent an inverse case to the loss of DNA methylation observed in some other species such as *Drosophila* and *C. elegans*.”.

Reviewer #3 (Remarks to the Author):

In this manuscript, authors describe the acquisition of cytosine methyltransferases by eukaryotic retrotransposable elements in two lineages: the dinoflagellate, *Symbiodinium kawagutii*, and a filamentous alga, *Klebsormidium nitens*. The detailed description of these kind of occurrences and their functional consequences for the retrotransposable element fitness and the genome would be very interesting, but I do not think the work gets there.

Authors should provide more details and analyses of the sequences to back up their claims and be cautious not to over interpret their results. It is OK to answer only a fraction of the questions and leave others unanswered. For example, I do not think the authors have shown yet that these acquisitions have the same effect in the different species or TEs, i.e., these acquisitions are an example of convergence, or are adaptations of the transposable elements (TEs) to the host epigenomic environment. I also think that authors should facilitate the access to all the genomic information.

Detailed comments:

Comment: 1. To facilitate access to the genomic information (TE copies and TE copies with CMT acquisitions) to readers/users not familiar with those genomes, links to the genome browsers for *Symbiodinium kawagutii* and *Klebsormidium nitens*, sequences and annotations of those sequences (TE copies and TE copies with CMT acquisitions) and a quick characterization of their distribution in the genome should be provided. Authors need to make easy for a reader to find those examples and understand their impact in the genome.

Response: We now include a link to all the data requested by the reviewer in this public repository: <https://figshare.com/s/8bf09ac32dcc1870c853>. The data include BED files with genomic locations of TE copies classified by TE type (DNMT, no DNMT, Copia, etc...) and by species, phylogenetic trees of the study and the original fasta files from which they were generated, annotation of the TEs that were cloned and used for the DNMT expression experiments, custom hmmer profiles generated for this study, a link to the original genome assemblies used and the annotation for *S. kawagutii* that we have generated using the newly generated RNA-seq data. The sequencing data generated for this study was already available in the GEO link provided in the text. Moreover, we have included the distribution of TE insertions in *S. kawagutii* as Supplementary figure 6a/b. We hope that this will allow readers to have easy access to this data, as it is all easily loaded into local genome Browsers such as IGV (software.broadinstitute.org/software/igv/), which is the browser that has been used to analyse and generate all the figures in this manuscript.

Comment: 2. Authors should also mention why they think that these TE copies with CMT acquisitions are not artifacts of the genome assemblies and provide details of any confirmatory results they have.

Response: We have done PCR and cloned full length copies of SymbioDIRS-Dnmt3 and SymbioLINE-Dnmt from genomic DNA of *S. kawagutii* (in order to express them in yeast as requested by Reviewer #4). We could obtain the whole PCR fragment in a single reaction for SymbioDIRS-Dnmt3 and bands that connect ORF1 and ORF2 for SymbioLINE-Dnmt.

Furthermore, finding the same structures hundreds of times in four independent genome assemblies due to artifacts of assembly is very unlikely, in particular when they are highly divergent in nucleotide composition. It would be also highly surprising to find such artifacts within a complex LTR such as DIRS, which has two repeats A1 and A2 flanking the TE

ORFs and two repeats B flanking the 3' repeat A2. Finally, DNMTs are in the same ORF as the reverse transcriptase and the RNase H in DIRS and Gypsy elements, and as the coding sequence is not repetitive it is less prone to misassemble. Finally, the *S. minutum* assembly is a product of Newbler assembler using a mix of 454 and Illumina data, while the rest of the genome assemblies are a mix of Illumina mate-pair libraries with different insert sizes. For the Illumina-only assemblies: *S. kawagutii* was assembled using SOAPdenovo, *S. microadriaticum* with ALLPATHS-LG and *S. goreau* with a mix of CLC Genomics Workbench, SPAdes95 and ALLPATHS-LG. Therefore, we do not think that all these different algorithms (and in the case of *S. minutum*, different sequencing technologies) would independently produce 3 different structures of DNMT-encoding retrotransposons independently by chance.

Comment: 3. I do not think the authors have shown yet that these acquisitions are “adaptations of the transposable elements to the host epigenomic environment” as claimed but it would be very interesting, if they were. At this point I only see one way to get that data. Authors could study the expansion of the different TE lineages in *Symbiodinium kawagutii* (i.e., the several SymbioDIRS structures with and without CMTs) and show what lineages have been more successful than others in the recent past (i.e., have produced more copies in the recent past). This would enhance the publication a lot. Conservation of the CMTs does not per se prove that they are beneficial for the TE transposable elements. Since no copy with introns of Dnmt3 is found, it could be that those are the source of Dnmt3 now (See point 5).

Response: We have now performed the suggested analysis using pairwise amino acid sequence identity to identify copies of TEs that have evolved recently in *Symbiodinium* genomes, revealing that DNMT and non-DNMT containing retrotransposons seem to have similar patterns of expansion (Supplementary figure 5a). This shows that having a DNMT is not a prerequisite to actively retrotranspose in *Symbiodinium* genomes. However, there are many mechanisms used by hosts to prevent retrotransposon activity, and TEs find ways to circumvent those mechanism using many strategies. For instance Krüppel-associated box genes are known to target some but not all TEs in vertebrates, and the targeted repetitive elements continue to escape from those silencing mechanisms. Therefore, acquiring a DNMT could still be an adaptation to the host epigenome. However, now we are more cautious about this interpretation and have rewritten the main text and abstract to accommodate hypothesis other than adaptation to the host epigenome.

Comment: Authors argue that “the fact that SymbioLINE-Dnmts carry CMTs that impose such a mutational burden upon themselves should only be tolerated if they confer beneficial effects to the retrotransposon.” The reality is that the retrocopy insertions of CMTs on TEs could make the fitness of the TE lower and we could still see copies as long as the fitness is not zero. So, a study of the recent success of the different TE lineages is the only way I see to test the TE fitness.

Response: We acknowledge the validity of this point by the reviewer and have now removed this statement from the revised manuscript.

Comment: 4. The Introduction about cytosine methyltransferases should be more general because different kinds of methylation have different effects and it is also genome

dependent (e.g., plants vs. animals) and authors discuss all of this later including a potential role related to RNA editing (Discussion).

This also relates to the alleged convergence. Authors observe CMTs in LTR and Non-LTR elements and they likely have different functional consequences for the different classes of TEs that have very different retrotransposition mechanisms and might not be examples of convergence.

Response: We have expanded the introduction on DNMTs and sequence/species specific roles of cytosine methylation.

We agree with the reviewer that DNMT likely have different roles/functions in different retrotransposons, therefore we now have removed all mentions to “convergent” evolution and highlight this point in the discussion. For instance, we have changed the title to “Recurrent acquisition of cytosine methyltransferases into eukaryotic retrotransposons”.

Comment: 5. The origin of the acquired CMTs is unclear (e.g., Dnmt3 of SymbioDIRS-Dnmt3). Where is the “parental” gene, i.e., gene with introns? Has it been lost? Is it a pseudogene or it is the portion of the genome that has not been sequenced? Have the retrocopies replaced that parental gene and this is why they are under purifying selection but they are not really beneficial for the TE? Or does the CMT come from a different species?

Response: The “parental” gene for SymbioDIRS-Dnmt3 seems to have been lost in *Symbiodinium*. However, multiple dinoflagellates species express Dnmt3 (without reverse transcriptases) in their transcriptomes (Figure 2f, Supplementary Figure 2). Moreover, SymbioDIRS-Dnmt3 sequences cluster well within the Dnmt3 group of eukaryotic sequences (which include plant, red algae and metazoan sequences). Therefore the most parsimonious scenario is that Dnmt3 was ancestrally found in dinoflagellates when it was recruited by retrotransposons and then it was subsequently lost as a host gene. We cannot identify Dnmt3 genes in the de novo transcriptome assemblies, which leads us to conclude that they have actually been lost and are not simply missing from the genome assembly. Given that most genes in *Symbiodinium* have very short exons (65 bp) pseudogenes should rapidly become undetectable.

The origin of SymbioLINE-Dnmt is unclear, as it is not reliably related to sequences from either eukaryotes, bacteria or viruses. Given the usual high substitution rates in dinoflagellates, it is likely that it has changed to such an extent (adapting to its new role as ORF1 in a LINE element) that its position cannot be assessed. More dinoflagellate genomes from other lineages might help to answer this. However, given the presence of a gene in a eukaryotic genome and not finding non-eukaryotic sequences branching close to it, the most parsimonious scenario is that it has been obtained vertically and not as product of Lateral Gene Transfer.

We now discuss this in the main text and also include the hypothesis of the DNMTs being beneficial for the host in the discussion. However, we do not think that the DNMT in the retrotransposons have replaced the ancestral Dnmt3 form. They are embedded in a Pol polyprotein together with a reverse transcriptase and an RNaseH domain, which could interfere with its function for host-related purposes. Furthermore, although some of the DNMT-encoding retrotransposons are transcribed, its transcription is biased, and in the case of DIRS, it does not include the ORF where the DNMT is found.

Comment: 6. Page 7. It is interesting that authors observe complete and not disabled copies of the TEs with acquired CMTs in different lineages. I am guessing they are comparing the consensus sequences but they should comment on this. It would be interesting to test if Ka/Ks between species for the same TE family that was acquired before the species split is significantly smaller than 1 although as mentioned in point 5 that might not be enough to show if these arrangements are beneficial for the TE but it would support that they have stayed functional even when the functional analyses in yeast are not conclusive.

Response: We have undertaken the analysis suggested by the reviewer which is now included in Supplementary Figure 8. We gathered sequences from all 4 *Symbiodinium* species belonging to lineages covering distant branches from Figures 2a and 2b. Then we obtained their closest orthologs from a *Symbiodinium* genome distinct from the query sequence and obtained the Ka/Ks values of the DNMT domain (see further details regarding how this analysis was conducted in the new Methods section “*Transposon similarity, sequence composition and selection analysis*”). In all cases the Ka/Ks ratio is < 0.2, which is strong evidence of purifying selection. But as suggested by the reviewer, we specify how purifying selection might be related to the host fitness and not only the retrotransposon’s own benefit.

Comment: 7. Page 12. There are aspects of the functional analyses of methylation that are not conclusive and authors should be clear about it. For example, there is a lack of correlation between methylation. TEs appear not to be silenced in all organisms (i.e., not in all the cells). Authors propose methylation occurs upon insertion but how does it end being so dynamic?

Response: We have rewritten this section so we do not claim that CH methylation silences transposons, just that it is found to be targeted to them.

We propose that methylation mediated by retrotransposon DNMTs must occur during the cDNA synthesis or upon insertion as the DNMTs are found in the Pol polyprotein together with the retrotranscriptase domain and the RNase H domain. In LINEs, ORF1 usually plays RNA binding roles and is co-expressed with ORF2, thus methylation should also be linked to RNA or newly synthesized cDNA upon insertion. However, this is just during the retrotransposition process. After insertion the retrotransposon ORFs are not expected to be transcribed at all times, and the methylation of the genomes (and on the transposons inserted in them) is most likely mediated by the host DNMT genes, which are abundant, highly expressed and belong to phylogenetically conserved families (Figure 1d, Supplementary Figure 2). However, *Symbiodinium* lacks Dnmt1, which is responsible in most eukaryotes for maintaining DNA methylation upon genome replication. Therefore we hypothesize that the noisy patterns of DNA methylation are due to the repertoire of not-well characterised Dnmt5 and Dnmt6 paralogs found in most dinoflagellates (Supplementary Figure 2).

Comment: 8. In several places in the text and figures authors assign expression to particular genes or TE copies (e.g., Page 3 or Page 8-Supplementary Figure 6). How do authors know what RNA-seq reads belong to the particular gene or copy depicted? Explain.

Response: Despite Kallisto being able to assign reads to multiple sequences (<https://www.nature.com/articles/nbt.3519>), it has been designed to discriminate isoform-

specific expression. Therefore, given that even closely related transposons accumulate mutations rapidly, as the whole *Symbiodinium* genome does, there are sufficient polymorphisms to allow unique mappability to discriminate between different copies. In *S. kawagutii*, among the 19 highest expressed transposable elements (>1 TPM) only 9 show nucleotide level identity above 90% to other transposable elements in the genome, and only two close relatives are both amongst the most highly expressed. These are the DIRS retrotransposons Skaw_blast2_1023 (8.7442 TPM) and Skaw_blast2_1471 (4.80869 TPM). Their nucleotide pairwise identity is 92% in the expressed region, however despite 2409 identical bases throughout their entire length, there are 229 SNPs distributed uniformly across the sequence that enable unique identification:

Furthermore, amongst the rest of the TEs with TPM >1 and copies with nucleotide pairwise identity >90, all their closest transposable elements show TPMs of 0 except one that is 0.0331955. Therefore we are confident that our measurements are accurate in terms of which copy is being expressed. Instead, if we required strict unique mappability of each read, we would be losing information that does not appear to add noise in the measurements. Moreover, we confirmed the expressed copies by blasting them against a de novo transcriptome assembly, where we confirmed that expression is positionally non-uniform, insofar that it not cover the whole transposable element sequence. As per reviewer’s 1 suggestion, we now have moved the expression levels figures to the Supplementary Materials and we do not draw major conclusions from the fact that individual transposons are transcribed or not, we simply explore whether retrotransposons of a certain type are expressed or not, confirming some transcription in all major classes (Supplementary Figure 9d).

Comment: 9. Page 19 – It is not clear what authors mean by “ubiquitous CG methylation in the *Symbiodinium* genome could be a response of the host to uncontrolled retrotransposon CMT methylation”. They should explain better this hypothesis.

Response: This is now developed as commented above in response to Reviewer #2. Please see the first paragraph of the revised discussion or the answer to Reviewer 2’s point 10.

Comment: 10. Page 21. Authors state “Many examples of domesticated transposable elements have been described across eukaryotes. These domesticated transposons acquire roles beneficial for the host ranging from transposon control, recruitment of cis-regulatory motifs or telomere elongation.” This should be rewritten. As it is currently written, it appears that the whole TE is domesticated but I think the authors mean TE protein domestication.

Response: This sentence has been rewritten accordingly: “Many examples of domesticated proteins with transposable element origin have been described across eukaryotes (Jangam et al. 2017).”

Reviewer #4 (Remarks to the Author):

The authors reported retrotransposons encoding cytosine methyltransferases (CMTs). These transposons were found in two eukaryotic lineages, dinoflagellates and charophytes. The authors also show that the genome of *Symbiodinium* have high level of cytosine methylation at CpG sites. Finally they show some of the transposon-encoded CMTs have activities to methylate CpG sites in the yeast.

Comment: In my opinion, it remains unclear how the transposon-encoded CMTs contribute for success of the transposons. The authors speculated that retrotransposons could self-methylate retro-transcribed DNA as a way to mimic host genomic DNA, but the evidence supporting that is limited. Without further evidences, their findings are rather descriptive. Overall, I feel the paper is too preliminary for publication in Nature Communications. Below are specific questions and comments, which might be useful for them.

Response: While we acknowledge that we cannot demonstrate the specific mechanism by which DNMTs are contributing to the success of the retrotransposons, our study contains many novel and exciting findings, as acknowledged by Reviewers 1, 2, 3 and 5. Among the novel findings that we report in this study we: 1) show how three types of retrotransposons have acquired DNMTs in *Symbiodinium*; 2) show how this happened independently in *K. nitens*; 3) profile for the first time the methylome of *Symbiodinium* species, revealing a unique methylome profile in eukaryotes amongst all those studied to date; 4) provide data for the methylome of *K. nitens*, the algal sister group to all land plants which is crucial to understand the evolution of cytosine methylation in this group; 5) characterise how these retrotransposons evolve and show that the DNMT domain is under purifying selection; 6) demonstrate the ability of some of the retrotransposon DNMTs to methylate in the CG context. Therefore, we think this manuscript has the breadth of novelty and data to be of interest to the broad audience of Nature Communications.

Comment: 1) Do unmethylated transposon copies silenced or excluded from the genomes of these species more efficiently than methylated copies? The authors should show that for supporting their speculation, because cytosine methylation function in the opposite way in vertebrates and plants; in these organisms, transposons are silenced by cytosine methylation.

Response: What the reviewer suggests is currently not feasible, as we can not experimentally test this hypothesis given that the host species are currently non-model systems and genetic manipulation tools are unavailable.

Nonetheless, we have now rewritten the manuscript and we discuss this amongst other hypotheses regarding why retrotransposons might have acquired cytosine DNMTs and the possible roles of cytosine methylation in the retrotransposition mechanism. As retrotransposons lacking a DNMT also show the ability to replicate in these genomes, it is possible that methylation is not used as a strict self-recognition mechanism. On the other hand, this mechanism might be functional but it can be circumvented by the non-DNMT encoding retrotransposons by using host DNMT genes to methylate their retrotranscribed cDNA.

Regarding the reviewer's comment on *Symbiodinium* and *K. nitens* methylomes having opposite functions to those of vertebrates and plants, we disagree. In *Symbiodinium* CH methylation clearly targets silent transposable elements, so it could also be a silencing mark in this species. Note that at this stage we do not know whether this is the case, and we do

not claim it in the new version of the manuscript. In the case of *K. nitens*, the methylome resembles that of land plants, with high levels of CHH and CHG methylation on transposable elements just like *Arabidopsis*, but with some interesting distinctions, such as gene body methylation that is more abundant in the 5' end of the gene rather than the 3'.

Comment: 2) When the full-length transposons are introduced into yeast or other species without genomic DNA methylation, do the CMTs self-methylate the transposons more efficiently than the other regions within the genome? Normally CMT can function in *trans*, and I do not understand why each of the transposons should keep the CMT genes.

Response: As suggested by the reviewer, we have now cloned full-length retrotransposons SymbioDIRS-Dnmt3-1, SymbioDIRS-Dnmt3-2 and SymbioLINE-Dnmt and expressed them in yeast under a galactose promoter, using the pPL2 backbone that has been used previously to express retrotransposons in yeast (<https://www.ncbi.nlm.nih.gov/pubmed/26104690>). After induction and growing the culture until stationary phase we extracted RNA and DNA from those cultures. We tested for transcription of the retrotransposon by qPCR and we sequenced the methylomes as described for the DNMT domain experiments.

We cannot detect any methylation induction by full-length retrotransposons, not even for SymbioDIRS-Dnmt3-1 and SymbioDIRS-Dnmt3-2, which we have shown to have DNMT activity when the domain is expressed in isolation. This is inconclusive, as retrotransposons co-evolve with their host genomes, and likely the signals required for translation in yeast are very different from those in *Symbiodinium*. *Symbiodinium* are known to require trans-splicing and show many divergent genomic features besides the DNA methylation patterns described in this manuscript. Given that the retrotransposons have several ORFs that have to be translated from the same transcript, the most likely scenario is that this does not happen properly in yeast as it has adapted to the peculiarities of *Symbiodinium*. As we think these results are inconclusive, we would prefer not to include them in the manuscript.

However, effectively DNMT can function in *trans*. In Figure 5b we show how the DNMT expressed in isolation can methylate both the yeast genome and the plasmid on which it is encoded. We show that the plasmid has higher methylation levels than the yeast genome, however, this is not a reliable indication of the DNMT acting preferentially in *cis* compared to

trans. First, because the Human DNMT3a DNMT domain shows the same pattern. Second, because it is potentially an effect of having many episomal copies of the plasmid in the nucleus that are probably not present in the same chromatin environment as the native genome, and are being sequenced at very different levels due to copy number (Supplementary Figure 14d).

However, in DIRS and Gypsy DNMT encoding retrotransposons, the DNMT is not in an isolated ORF. The DNMT is in the same ORF as the reverse transcriptase domain and the RNase H domain. Therefore, when this multiprotein is translated, it is highly likely that methylation happens at the same time as retrotranscription and degradation of the RNA:DNA heteroduplex. Therefore, due to the multi-domain architecture of these proteins with basic roles in retrotransposition it is expected to work in *cis*, only on the retrotranscribed cDNA produced by the reverse transcriptase.

Comment: 3) An alternative possibility for the presence of CMTs in the transposons could be silencing of endogenous CMTs by RNAi? If so, host CMTs evolve rapidly to escape from the silencing by the transposon-encoded CMTs and the transposons incorporate the new CMTs constantly from the host genome to silence them. Do the transposon-encoded CMTs evolve rapidly in the nucleotide sequence level? Are small RNAs for CMTs detected in these species?

Response: We do not find evidence for this scenario, as all *Symbiodinium* DNMT-encoding retrotransposons seem to have ancient roots. DNMTs were acquired 3 times by retrotransposons in this lineage, and maintained from then on. Moreover, transposon-encoded DNMTs show evidence for purifying selection, which suggests a pressure to be conserved instead of the opposite. In fact, the scenario proposed by the reviewer would require many assumptions that are currently unknown or not supported by the current data, such as transposon-encoded DNMT being able to silence host DNMTs. Thus, given the current data, we do not envisage a mechanism by which this could happen. Moreover, CG methylation is widespread in the genome, including gene promoters, and therefore not likely involved in silencing. Regarding CH methylation, which could be involved in silencing due to its patterns of deposition on transposable elements, the yeast DNMT assays did not demonstrate the ability of retrotransposon-encoded DNMTs to methylate in this context, and we do not find CH methylation enriched in host DNMT genes. Therefore, we do not think this is a likely scenario based on our data.

Reviewer #5 (Remarks to the Author):

The manuscript by de Mendosa et al. reports a thorough analysis of retrotransposon-associated methyltransferase domains in poorly studied taxa, dinoflagellates and charophytes. It is well-written, technically solid, and provides a comprehensive inventory of MTases, as well as the data on transcription, bisulfite sequencing, and activity of selected MTases in a heterologous system.

Comment: While the subject is certainly of interest, the MS nevertheless leaves the reader unsure what the data actually means, and whether it should be interpreted in the way it is presented in the abstract. To promote the presented concept that retrotransposons acquire MT genes repeatedly to mimic themselves as host genomic DNA, the authors need to

demonstrate the existence of host response to unmodified genomic invaders in this particular host.

Response: As mentioned above in response to Reviewer 4 (comment #1), current techniques do not allow genetic manipulation of the host species, making these experiments unfeasible at this stage. However, we have reworded the manuscript so this hypothesis is among other possible explanations.

Comment: Barring that, the null hypothesis could be that the transposon-encoded MT may be just decorating DNA with little or no consequences for either the element or the host. Could some MTs, including the ones from obscure clades, be circulating in the aquatic environment, e.g. in viruses, and be repeatedly captured from such sources by horizontal means, and eventually lost?

Response: Given the signal for purifying selection on the DNMT domain (Supplementary figures 8 and 12) and the evolutionary age of some of these retrotransposons (Figure 4e), Our data indicates that the non-adaptive presence of DNMTs is the least parsimonious scenario. Although we cannot at this stage unambiguously demonstrate the function of these DNMTs, it is quite well supported that they must have a function, which is the premise for purifying selection, and also considering the conservation of key functional amino acid positions, the activity shown in yeast, and the patterns of evolutionary expansion of these retrotransposons.

According to Reviewer's suggestion, we have blasted the retrotransposon-DNMT domains against viral sequences from NCBI and incorporated them to the phylogenetic analysis (we have revised Figure 1f, Figure 4a and Supplementary Figure 2). In all cases, viral sequences did not cluster close to the retrotransposon DNMTs, which is a strong indication that they are not related. Moreover, SymbioDIRS-Dnmt3 sequences cluster within the eukaryotic clade of DNMT3 with robust nodal support, including DRM from plants and DNMT3 from other dinoflagellate transcriptomes. Even if they were acquired from a not yet sequenced virus, currently little is known about specific mechanisms by which viral DNMTs function (<https://www.ncbi.nlm.nih.gov/pmc/articles/PMC2396429/>). Finally, the phylogenetic tree of DNMTs do not support many repeated acquisitions, but rather 3 in *Symbiodinium* and 1 in *K. nitens*.

Comment: It would be good to see convincing evidence that they were indeed captured from the host, as the MS claims. In Fig. 1, this information is not made visible, i.e. where the intron-containing MTs are placed on the tree - only the domain structure is presented in Fig. S1. Are RT and MT domain topologies fully congruent? Maybe they can share/exchange components?

Response: We apologize for not having clarified Figure 1f in the previous version of the manuscript. In the revised manuscript we now clearly indicate which branches belong to exon-rich genes and which belong to mono-exonic genes.

As mentioned in the previous response, retrotransposon-Dnmt3 sequences always cluster within the eukaryotic clade of Dnmt3 sequences. This, and the presence of Dnmt3 in the transcriptomes of other dinoflagellates (Figure 1f, Supplementary figure 2), indicates that Dnmt3 was not only ancestral to eukaryotes but was also found in the last common ancestor of dinoflagellates. This makes the hypothesis of vertical descent and subsequent loss of the

native Dnmt3 in the lineage that gave rise to *Symbiodinium* more likely than Lateral Gene Transfer. The origin of the DNMT in SymbioLINE-Dnmts is less clear, as it is quite divergent from any other clade of Dnmt. Given that neither bacterial, viral or eukaryotic sequences cluster reliably nearby, the most parsimonious scenario is that some copy of Dnmt encoded by the host was acquired by a LINE element and underwent a process of rapid evolution that masks its deep phylogenetic origin. The alternative, suggesting Lateral Gene Transfer from an unknown donor in a eukaryote, would be much more difficult to argue due to the current knowledge on eukaryote Lateral Gene Transfer. Despite there being good examples of genes acquired through LGTs in eukaryotes, this mechanism should never be taken as the null hypothesis given the challenges in explaining the mechanisms of acquisition of foreign DNA in eukaryotes, and the generally more parsimonious scenarios explained by pervasive secondary gene loss resulting in genes with patchy distribution. Moreover, in the case of *K. nitens*, the native Dnmt3 and the retrotransposon-Dnmt3 coexist in the same genome, thus there is not doubt in this case.

According to reviewer's suggestion, we have now included a tree of retrotransposon-DNMTs (Supplementary figure 7), which shows that the reverse transcriptase and the DNMT domain phylogenies are congruent, as the three *Symbiodinium* DNMT-encoding retrotransposons form monophyletic clades, distinguishing SymbioLINE-Dnmts from SymbioDIRS-Dnmt3, and moreover, structure A and B of SymbioDIRS-Dnmt3 are sister clades. This could either mean that acquisition in each structure happened independently, or in the potential case that one structure acquired the Dnmt3 from the other, it has diverged substantially since then.

Comment: Also, if the profiles for CG and CH differ, different enzymes would likely be responsible for modification – since DIRS MTs favor CG, perhaps one of the host MTases is at work?

Response: We totally agree with the reviewer. In fact the host transcribes more exon-rich DNMT genes than humans or plants (several dinoflagellate-specific paralogs of Dnmt5 and Dnmt6, as shown in figure 2d and Supplementary figure 2), which clearly indicates that they must have an active role in the host methylation. We do not think that retrotransposon DNMTs are the main source of methylation in *Symbiodinium* (or *K. nitens*) genomes due to their low expression levels, their presence in multi-domain ORFs and likely confined translation in episodic retrotransposition events. But we think that their presence might have an influence upon the methylome composition. We now try to make this point more clear in the Discussion: "...the methylation patterns observed in *Symbiodinium* genomes are likely deposited by the several copies of highly expressed exon-rich DNMTs belonging to Dnmt5 and Dnmt6 families, as in other algal lineages where Dnmt1 and Dnmt3 are absent".

Specific comments:

Comment: -No analysis of RT-MT retrotransposons in the published *Symbiodinium microadriaticum* genome (Sci Rep 2016, 6:39734) is presented. It is reported to be ancestral to *S. minutum* and *S. kawagutii*, and to contain ~2700 copies of DIRS elements, some of which are MT-containing. Does it show similar CG under-representation in those transposons?

Response: We have now added data and analysis for the retrotransposons found in *S. microadriaticum* and *S. goreau* (which has been recently made available). We find

SymbioLINE-Dnmt and both structures of SymbioDIRS-Dnmt3 in both genomes, suggesting that those retrotransposons were ancestrally present in the Last Common Ancestor of *Symbiodinium* species. Furthermore, in all 4 species of *Symbiodinium* we observe the depletion of CG dinucleotides in SymbioLINE-Dnmts compared to LINE elements without DNMT domain (Figure 2d).

Comment: -The presence of MT in DIRS elements has been known for a long time, and the list of references to MT-encoding DIRS could be significantly expanded (e.g. fungi, nematodes): Muszevska et al. PLOS One 2013, 8(9); Lunt and coworkers PLOS One 2014, 9(9).

Response: We agree, and in fact, the presence of a DNMT domain in DIRS was reported in the first description of DIRS elements (Goodwin & Poulter, Mol Biol Evol, 2004). However, those are adenine DNMTs, which are structurally unrelated to cytosine DNMTs, and are more closely related to bacterial DNA Adenine Methyltransferase (Dam) DNMTs. Given that Dam-like DNMTs are scarce in eukaryotes (at least as “native” host genes) it is less likely that it represents a similar case of acquisition from the host. Moreover, given that Adenine DNMTs are found in DIRS elements of such divergent clades (*Dictyostelium*, deuterostomes, nematodes, *Chlamydomonas*, fungi) it is more parsimonious to hypothesize that it was ancestrally found in some ancestral form and inherited vertically. Another example of common descent would be the Chromo domain found in the Chromoviruses of Gypsy families (<https://www.ncbi.nlm.nih.gov/pubmed/20377908>). These domains acquired by families that have such deep origins makes it difficult to test whether they were acquired by the transposable elements or whether the eukaryotes took these domains from the transposable elements.

We now report this difference in the results and in the discussion of the manuscript, including the references suggested by the reviewer.

Comment: -Convergent evolution of other components, e.g. RNase H repeatedly captured by different retrotransposons (L1, gypsy), in many of them several times, was reported in diverse taxa, such as oomycetes, plants, and fungi (PNAS 2013, 110:20140; Mob DNA, 2017, 8:4; Mol Biol Evol 2015, 32:1197).

Response: These past studies are now included in the discussion: “There are several reports of recurrent acquisition of protein domains by retrotransposons (RNase H, Chromo domain), mostly recruited from other transposable elements but also from host genes (Ustyantsev et al. 2015; Smyshlyaev et al. 2013; Ustyantsev et al. 2017).”.

Reviewers' comments:

Reviewer #1 (Remarks to the Author):

I am satisfied that the authors have addressed my comments.

Reviewer #2 (Remarks to the Author):

The reviewers have largely satisfactorily addressed my comments.

On line 105, the authors state that intronless DNMT families are not retrocopies of transcribed genes. How do they know that? It's not clear to me why it matters to their overall argument that they are not retrocopies.

The authors argue that reduced CG dinucleotides in SymbioLINE-Dnmts would be expected if they self-methylate. It would also be expected if SymbioLINE-Dnmts are simply more likely to be methylated by other DNMTs in the genome than are the LINEs lacking DNMT domains. It doesn't seem that the authors can pinpoint the DNMT that is the source of the methylation based on this analysis.

Reviewer #3 (Remarks to the Author):

Authors have now answered only a fraction of my comments. There are two main questions that I do not think have been addressed properly.

Related to my comment 3, authors do not explain well how they study the expansion dynamics. Authors say in their answer "We have now performed the suggested analysis using pairwise amino acid sequence identity to identify copies of TEs that have evolved recently in Symbiodinium genomes, revealing that DNMT and non-DNMT containing retrotransposons seem to have similar patterns of expansion (Supplementary figure 5a)." I do not see details of the whole dynamics for the copy numbers of these retrotransposons in this figure or Supplementary figure 5c. Because density is given per kind of element and not comparing Dnmt with nDnmt in a genome and their changes in copy numbers, it is unclear if authors can argue that the acquisition of Dnmt increased the retrotransposon fitness in the discussion unless Dnmt numbers quickly increased compared to nDnmt retrotransposons at some point in time.

Related to my comment 6, I suggested that authors sample copies within a lineage (i.e., recent insertions), built the consensus and compare that sequence to the consensus of recent insertions within another lineage to infer how this gene is evolving within TEs. This is compare consensus sequences for recent copies between lineages. For comparison, they could do the same for other TE proteins that have stayed functional as retrotransposons are able to produce copies of themselves and they should find purifying selection for all of them. They should not compare to an orthologous gene.

Reviewer #4 (Remarks to the Author):

It is reasonable that in the revised manuscript the authors have toned down their original claims about contribution of the TE-encoded DNMTs for adaptation of the TEs. No evidence supports the hypothesis that methylation helps expansion of the TEs (the authors' responses to comment 3 of reviewer 3,

comment 1 of reviewer 4, and comment 1 of reviewer 5). The manuscript is technically fine. If the editor regards that their observations are exciting enough to justify publication in this journal, I do not have a strong objection to that.

Reviewer #5 (Remarks to the Author):

This thoroughly revised version was much improved by dropping the unsupported claims from the abstract and the results (e.g. use of methylation to discriminate between host and transposon DNA, or the utility of transposon-encoded Dnmts). The authors now propose a different explanation, i.e. they speculate that after Dnmt acquisition by transposons and its loss by the host cells, the host switched to the all-methylated state as a default, to override any regulatory mechanisms that could have resulted from differential methylation patterns. Overall, the focus seems to be shifting from transposon-borne Dnmts to unusual methylation patterns in the host, likely augmented by the presence of Dnmt5 and Dnmt6 genes, but the significance of Dnmt capture by transposons is still unclear. Domestication usually refers to recruitment of transposon components outside of transposon boundaries, i.e. immobilization in the genome.

I am still not fully convinced that unknown Dnmt-rich viruses circulating in marine organisms can be eliminated from the picture as potential vehicles and sources for lateral gene exchanges. Independent acquisition of DNMT genes has been previously described in DNA transposons inserted into large DNA viruses of marine algae: Mariner-1_OLpv and Mariner-2_PGv (Repbase) (Bao and Jurka 2013, *Mob DNA* 4:12, Fig. 3). Those authors did not offer hypotheses regarding the roles possibly played by Dnmts in the transposition cycle; the present MS does not shed much light on these roles either. The presence or absence of Dnmt in dinoflagellate or charophyte transposons does not seem to influence either proliferative capacity or transcriptional activity of the transposons. Thus, the concluding statement in the abstract "Together, this is an example of how retrotransposons incorporate host-derived genes involved in DNA methylation may be linked to the composition and regulation of the host epigenomic environment", which is by itself an awkward sentence and needs rewording, may disorient the readers into thinking that such a connection has been revealed.

Reviewers' comments:

Reviewer #1 (Remarks to the Author):

I am satisfied that the authors have addressed my comments.

Reviewer #2 (Remarks to the Author):

The reviewers have largely satisfactorily addressed my comments.

Comment: On line 105, the authors state that intronless DNMT families are not retrocopies of transcribed genes. How do they know that? It's not clear to me why it matters to their overall argument that they are not retrocopies.

Response: We know they are not retrocopies of host genes as they are found within retrotransposon structures (either LTR or LINE typical features). Retrogenes are host genes that got retrotransposed back into the genome by reverse transcriptase by accident, thus they lack introns, but they don't have the capacity to retrotranspose as they are not part of a retrotransposon. However, if they are within a retrotransposon coding sequence it means that they must be used by the retrotransposon. Most likely, the acquisition of DNMTs by retrotransposons was mediated by fusion of a host gene retrocopy to a retrotransposon, but from then on, the new copies of the intronless DNMT expanded due to being part of the TE, and not as an accidental retrotransposition of host transcribed genes. Therefore, intronless DNMT being found within retrotransposons and not just as accidental retrocopies of host genes is central to the manuscript, as this is what allowed the dramatic expansion of this gene family in both *Symbiodinium* and *K. nitens* genomes.

Comment: The authors argue that reduced CG dinucleotides in SymbioLINE-Dnmts would be expected if they self-methylate. It would also be expected if SymbioLINE-Dnmts are simply more likely to be methylated by other DNMTs in the genome than are the LINES lacking DNMT domains. It doesn't seem that the authors can pinpoint the DNMT that is the source of the methylation based on this analysis.

Response: We have now included this possible explanation in the manuscript.

Reviewer #3 (Remarks to the Author):

Authors have now answered only a fraction of my comments. There are two main questions that I do not think have been addressed properly.

Comment: Related to my comment 3, authors do not explain well how they study the expansion dynamics. Authors say in their answer “We have now performed the suggested analysis using pairwise amino acid sequence identity to identify copies of TEs that have evolved recently in *Symbiodinium* genomes, revealing that DNMT and non-DNMT containing retrotransposons seem to have similar patterns of expansion (Supplementary figure 5a).” I do not see details of the whole dynamics for the copy numbers of these retrotransposons in this figure or Supplementary figure 5c. Because density is given per kind of element and not comparing Dnmt with nDnmt in a genome and their changes in copy numbers, it is unclear if authors can argue that the acquisition of Dnmt increased the retrotransposon fitness in the discussion unless Dnmt numbers quickly increased compared to nDnmt retrotransposons at some point in time.

Response: We apologize if our analysis did not fully address the point raised by the reviewer in the previous revision, but as far as we understood the reviewer’s concerns, our approach answered to some extent these points, which we will expand upon further here. In the analysis we first set a threshold of amino acid identity (80% identical residues) among transposable elements to differentiate which are “recent copies” (having very similar copies in the same genome implies that they have diverged relatively recently) versus “old copies” (not having closely related family members in the genome may imply that those are older copies that are no longer active). Our reasoning on focusing on “recent copies” was to investigate whether TEs containing Dnmts were able to replicate faster than other TE classes that don’t possess DNMTs in *Symbiodinium* genomes. Focusing on recent events would also correct for evolutionary contingencies such as Copia elements being the most abundant in these genomes, therefore even if they were less likely to retrotranspose there would be more total copies, which might not be able to replicate anymore, just by historical reasons. But in our analysis this was not what we observed (see Supplementary Figure 5a). In fact, the recent copies are mostly Copia LTRs, indicating that recent transposition events just reflect the relative global number of TEs in the genome, and all types of retrotransposons are able to retrotranscribe (Copia, DIRS without DNMT, DIRS with DNMT, LINE without DNMT and LINE with DNMT). This is true for total copy number as well as for total number of TE subfamilies (e.g. collapsing all 80% identical TEs into a single number, as there are many different subfamilies expanding, not just a single subfamily). Consequently, we had already mentioned this lack of differential dynamics among retrotransposon types as it reads in the Results section: “*When restricting the search for recent copies, all types of retrotransposons show similar expansion dynamics, some new and actively expanding lineages but a majority of divergent lineages (Supplementary Fig. 5a,c). This indicates that encoding a DNMT is not required in order to be retro-transcriptionally active in Symbiodinium genomes.*”, as well as in the Discussion: “*As DNA methylation is specifically interpreted depending on its base, sequence context and genomic location, the methylation status of the newly retrotranscribed copies of DNMT-encoding retrotransposons may affect the chromatin configuration of the insertion site. Nevertheless, given the ability of non-DNMT retrotransposons to retrotranspose in these genomes, these hypothetical defense mechanisms that the host might have to target unmethylated DNA can be circumvented by other mechanisms other than encoding a DNMT.*” Therefore, we are not claiming that DNMT-encoding retrotransposons are more fit than non-DNMT-encoding

TEs. Rather, we just claim that recurrent acquisition of DNMT domain into 3 different lineages of retrotransposons, conservation of this domain (key amino acids and Ka/Ks ratios < 0.2), conservation of the retrotransposon superfamilies during more than 50 million years of *Symbiodinium* evolution, and relatively high numbers of DNMT-encoding subfamilies support that this is not a neutral event. Encoding a DNMT must have fitness implications for the retrotransposons encoding them, otherwise these retrotransposons would be found in very low numbers, would not show sequence conservation, and DNMTs would not have been acquired by 3 independent lineages of retrotransposons.

Nonetheless, as suggested by the reviewer, we have now performed a new analysis (Supplementary Figure 5c) focusing on the gain and loss of specific transposable element subfamilies across *Symbiodinium* evolution divided by DNMT and non-DNMT transposable elements. First we defined subfamilies using 60% amino acid identity threshold in a blastp search (using 80% amino acid identity retrieved only species-specific subfamilies, thus we relaxed this criterion to find subfamilies spanning more than one species) and used identity values as edges in a similarity network to define modules of connected sequences as subfamilies/orthologous groups (as in Supplementary Figure 5a). Then we built a count matrix for each subfamily in each species, which we have implemented into Count (<https://www.ncbi.nlm.nih.gov/pubmed/20551134>), a software designed to reconstruct gene gains, losses and inferring ancestral node reconstructions. We performed the analysis in separate for DNMT containing subfamilies (including both SymbioLINE-Dnmt and SymbioDIRS-Dnmt3 superfamilies) and for subfamilies of non-DNMT encoding transposable elements (DIRS, LINE, Copia, DNA transposons).

Consistent with our previous approach, we find that both DNMT and non-DNMT encoding retrotransposons are able to replicate (and thus generate new subfamilies) in *Symbiodinium* genomes. Finding many different types of retrotransposons active in the same genome is not surprising given that there is not a single mechanism to silence TEs, moreover different TE families use different strategies to avoid silencing mechanisms. For instance, Alu, L1 and HERV are all active in the human genome, using different mechanisms to thrive. Thus, we cannot conclude, as the reviewer suggests, that DNMT encoding retrotransposons increased retrotransposon fitness just by comparing gains and losses to non-DNMT encoding transposons. But as we were not arguing that in the discussion section of the manuscript, we

did not implement any changes in the text but in Methods section explaining this new analysis.

Comment: Related to my comment 6, I suggested that authors sample copies within a lineage (i.e., recent insertions), built the consensus and compare that sequence to the consensus of recent insertions within another lineage to infer how this gene is evolving within TEs. This is compare consensus sequences for recent copies between lineages. For comparison, they could do the same for other TE proteins that have stayed functional as retrotransposons are able to produce copies of themselves and they should find purifying selection for all of them.

They should not compare to an orthologous gene.

Response: We apologise if we misunderstood the reviewer's previous suggestion. To address the reviewer's point, we have now included all vs all Ka/Ks ratios intra-species comparisons for DNMT and reverse transcriptase domains (collapsing identical copies) for each retrotransposon type. The results can be seen in the new Supplementary Figure 8. Interestingly, most comparisons show Ka/Ks ratios below 0.2 (as reported in the inter-species comparison). In most cases the Ks ratio (synonymous mutations) is saturated, which clearly indicates that there are a lot of mutations (as previously mentioned, *Symbiodinium* genomes evolve very fast compared to vertebrates or plants), but that active purifying selection is preserving the DNMT domain. Even in those comparisons where Ks is not saturated, we observe signal for purifying selection. In a couple of cases, we detect TEs that are not under purifying selection, rapidly accumulating non synonymous mutations at the same rate as synonymous in both DNMT and RT domains, suggesting that they are inactive. When we compared the Ka/Ks ratios of the DNMT and the reverse transcriptase found in the same transposable elements, their ratio of Ka/Ks ratios were very close to 1, indicating that both domains are conserved at similar levels (see new Supplementary Figure 9a), thus likely equally important for the retrotransposon.

Reviewer #4 (Remarks to the Author):

It is reasonable that in the revised manuscript the authors have toned down their original claims about contribution of the TE-encoded DNMTs for adaptation of the TEs. No evidence supports the hypothesis that methylation helps expansion of the TEs (the authors' responses to comment 3 of reviewer 3, comment 1 of reviewer 4, and comment 1 of reviewer 5). The manuscript is technically fine. If the editor regards that their observations are exciting enough to justify publication in this journal, I do not have a strong objection to that.

Reviewer #5 (Remarks to the Author):

Comment: This thoroughly revised version was much improved by dropping the unsupported claims from the abstract and the results (e.g. use of methylation to discriminate between host and transposon DNA, or the utility of transposon-encoded Dnmts). The authors now propose a different explanation, i.e. they speculate that after Dnmt acquisition by transposons and its loss by the host cells, the host switched to the all-methylated state as a default, to override any regulatory mechanisms that could have resulted from differential

methylation patterns. Overall, the focus seems to be shifting from transposon-borne Dnmts to unusual methylation patterns in the host, likely augmented by the presence of Dnmt5 and Dnmt6 genes, but the significance of Dnmt capture by transposons is still unclear. Domestication usually refers to recruitment of transposon components outside of transposon boundaries, i.e. immobilization in the genome.

Response: In the manuscript we overtly emphasize that the biological significance of recurrent of Dnmt acquisition by different retrotransposons and its link to the unique epigenomic characteristics of the host is still unclear, as currently there is no way to test this experimentally. Regarding the use of the word “domestication”, we only use it 3 times in the manuscript, and in all cases we are referring to proteins derived from transposable elements being recruited by the host genome. Therefore, we did not implement any changes in the text regarding the use of this word.

Comment: I am still not fully convinced that unknown Dnmt-rich viruses circulating in marine organisms can be eliminated from the picture as potential vehicles and sources for lateral gene exchanges. Independent acquisition of DNMT genes has been previously described in DNA transposons inserted into large DNA viruses of marine algae: Mariner-1_OLpv and Mariner-2_PGv (Rebase) (Bao and Jurka 2013, *Mob DNA* 4:12, Fig. 3). Those authors did not offer hypotheses regarding the roles possibly played by Dnmts in the transposition cycle; the present MS does not shed much light on these roles either.

Response: As the reviewer suggests, it cannot be ruled out that viruses were the origin of the DNMTs, as the absence of evidence is not evidence of absence, thus it could always be argued that an unidentified organism might be the source of these domains. However, none of the current evidence points to this conclusion. To address the reviewer’s specific point, we have analyzed Mariner-1_OLpv and Mariner-2_PGv sequences, and found that only Mariner-1_OLpv has a bona fide cytosine DNMT, while Mariner-2 encodes a AdoMet-MTase, which is a distantly related family of enzymes. We used Mariner-1_OLpv DNMT sequence in the following phylogenetic tree (in blue eukaryotic DNMT clades, in pink, retrotransposon DNMT, in grey, bacterial/viral clades, in black, bacterial/viral branches):

Therefore, not only is Mariner-1_OLpv a DNA transposon (not a retrotransposon) found in a giant virus genome (and not in a eukaryote genome), but its sequence does not cluster close to any of the described DNMT types found in the retrotransposons described in this manuscript, but rather with bacterial and viral sequences. Therefore invoking a yet to discover virus as an intermediate step on the DNMT domain acquisition by retrotransposons instead of the more parsimonious vertical acquisition seems a very unlikely explanation for the origin of these domains. Moreover, giant virus are not yet known to infect *Symbiodinium*, although it would be likely given the emerging evidence in many eukaryotic lineages. Nevertheless, now we have included a sentence in the manuscript to discuss this possibility in the case those unknown viruses get sequenced in the future. Additionally, we cite the paper mentioned by the reviewer.

Comment: The presence or absence of Dnmt in dinoflagellate or charophyte transposons does not seem to influence either proliferative capacity or transcriptional activity of the transposons. Thus, the concluding statement in the abstract "Together, this is an example of how retrotransposons incorporate host-derived genes involved in DNA methylation may be linked to the composition and regulation of the host epigenomic environment", which is by itself an awkward sentence and needs rewording, may disorient the readers into thinking that such a connection has been revealed.

Response: We have rephrased the sentence in the abstract avoiding possible confusion to readers, avoiding any possible interpretation regarding the proliferative or transcriptional activities of DNMT encoding retrotransposons.